# Optimistically Tempered Online Learning

## Abstract

Optimistic Online Learning algorithms have been developed to exploit expert advices assumed optimistically to be always useful. However, it is legitimate to question the relevance of such advices *w.r.t.* the learning information provided by gradient-based online algorithms. We develop in this work the *optimistically tempered* (OT) online learning framework as well as OT adaptations of online algorithms. Our algorithms come with sound theoretical guarantees in the form of dynamic regret bounds and we eventually provide experimental validation of the usefulness of the OT approach.

## 1 Introduction

Online learning (OL) is a paradigm in which data is processed sequentially, either because the practitionner does not collect all data prior to analysis or because the dataset dynamically evolves through time, or simply because handling batch of data is numerically too demanding. From the seminal work of Zinkevich (2003), which proposed an online version of the celebrated gradient descent algorithm, OL has been at the core of many contributions (we refer to Hazan et al., 2007; Duchi et al., 2011; Rakhlin and Sridharan, 2013a for an overview). The classical performance criterion of an online learning algorithm is the *static regret*. Given a sequence of loss functions $(\ell_t : \mathcal{K} \to \mathbb{R})_{t \geq 1}$, the static regret compares the efficiency of a sequence of predictors $\hat{\mu} = (\hat{\mu}_t)_{t \geq 1}$ to the best fixed strategy: S-Regret$_T(\hat{\mu}) = \sum_{t=1}^{T} \ell_t(\hat{\mu}_t) - \inf_{\mu_0 \in \mathcal{K}} \sum_{t=1}^{T} \ell_t(\mu_0)$, $T > 0$. Classical upper bounds on static regret involve a sub-linear rate. For instance, Zinkevich (2003) proposed a $\mathcal{O}(\sqrt{T})$ bound for Online Gradient Descent (OGD) which is valid for convex losses. Hazan et al. (2007) proved a $\mathcal{O}(d \log(T))$ rate for the Online Newton Step (ONS) algorithm with exp-concave losses when $\mathcal{K} \subseteq \mathbb{R}^d$.

**Dynamic Regret.** Static regret may not be sufficient to assert the efficiency of an online algorithm as the class of static strategies is limited compared to all possible strategies. Hence the notion of *dynamic regret* introduced by Zinkevich (2003) and further developed by Hall and Willett (2013). For any sequence $\hat{\mu}$ of predictors and any sequence $\mu$ of dynamic strategies, the dynamic regret is given by

$$\text{D-Regret}_T(\hat{\mu}, \mu) = \sum_{t=1}^{T} \ell_t(\hat{\mu}_t) - \sum_{t=1}^{T} \ell_t(\mu_t), \qquad T \geq 1 \,.$$

Dynamic regret has attracted many studies recently, especially when the comparator sequence is $\mu^* = (\mu_t^*) = (\inf_{\mu \in \mathcal{K}} \ell_t(\mu))_{t \geq 1}$ (worst-case dynamic regret, as in Besbes et al., 2015; Jadbabaie et al., 2015; Yang et al., 2016; Zhang et al., 2017; 2018b; Zhao and Zhang, 2021) but also for any comparator sequence (universal dynamic regret, as in Zhao et al., 2020). Those works have established various upper bounds which depend on measures of the cumulative distance between successive optima. For any horizon $T \geq 1$, for any sequence $\mu = (\mu_t)_{t \geq 1}$, Zinkevich (2003) introduced the *path length* to measure this discrepancy $P_T(\mu) = \sum_{t=1}^{T-1} \|\mu_{t+1} - \mu_t\|$. Zhang et al. (2017) introduced the *squared path length*: $S_T(\mu) = \sum_{t=1}^{T-1} \|\mu_{t+1} - \mu_t\|^2$. Finally, the *function variation* has been introduced by Besbes et al. (2015): for any sequence of losses $(\ell_t)_{t \geq 1}$ (these are provided by the environment),$V_T^\ell(\mu) = \sum_{t=1}^{T-1} \sup_{\mu \in \mathcal{K}} \|\ell_{t+1}(\mu) - \ell_t(\mu)\|$. When using the path length[1] $P_T^* := P_T(\mu^*)$ of the minimisers $\mu^* = (\mu_t^*)_{t \geq 1}$, dynamic regret of OGD is at most $\mathcal{O}(\sqrt{T(1 + P_T^*)})$ for convex functions

---

[1]similar definitions hold for the squared path length and the function variation.

(Zinkevich, 2003; Yang et al., 2016). We similarly define $S_T^* := S_T(\mu^*)$.

For strongly convex and smooth functions, Mokhtari et al. (2016) established that the dynamic regret is $\mathcal{O}(P_T^*)$. Zhang et al. (2017) introduced the Online Multiple Gradient Descent (OMGD) and the Online Multiple Newton Update (OMNU) which achieved a $\mathcal{O}(\min(P_T^*, S_T^*))$ dynamic regret. Yang et al. (2016) showed that the $\mathcal{O}(P_T^*)$ rate is also reached for convex and smooth functions under the assumption that all minimisers lie onto the interior of a convex set of interest. Besbes et al. (2015) proved a $\mathcal{O}(T^{2/3}(V_T^*)^{1/3})$ dynamic regret for OGD with a restarting strategy. Finally, Baby and Wang (2019) improved the rate to $\mathcal{O}(T^{1/3}(V_T^*)^{2/3})$ for 1-dimensional square loss with filtering techniques. Note that all the aforementioned results assume implicitly access to $P_T^*$, $S_T^*$, $V_T^*$ and that a notion of *universal dynamic regret* has been studied by Zhang et al. (2018a); Zhao et al. (2020; 2022) to compete with any $P_T(\mu)$, $S_T(\mu)$, $V_T(\mu)$ rather than $P_T^*$, $S_T^*$, $V_T^*$.

**Optimistic online learning (O-OL).** Optimistic online learning exploits, at each time step, a (possibly) history-dependent additional information provided by an expert. Being optimistic in this context is relying on the fact that the expert advices are relevant and can be exploited within an optimization procedure. Optimistic online learning can be traced back to Hazan and Kale (2010); Chiang et al. (2012) and has been further developed by Rakhlin and Sridharan (2013a;b) which introduced the celebrated Optimistic Mirror Descent (OptMD). Those works involved static regret bound exploiting explicitly the experts' advice. Jadbabaie et al. (2015) bridged the gap between dynamic regret and optimistic online learning by providing an adaptive version of OptMD allowing to obtain dynamic regret bounds for bounded convex functions.

## 1.1 A general class of online algorithms with expert advices.

O-OL algorithms rely on a trust in available expert advice, which is, for instance, directly incorporated in the dual space for the OptMD algorithm. More generally, in what follows, we consider the class of *gradient-based online learning (GB-OL) algorithms with judge $f$* whom the update phase consists in a gradient step alongside the incorporation of additional knowledge through the judge $f$. With expert advice $\nu$ (being a sequence of vectors in $\mathbb{R}^d$), a GB-OL algorithm satisfies the pattern of Algorithm 1.

---

**Algorithm 1:** A GB-OL algorithm with judge $f$.

**Parameters** : Horizon $T$, step-sizes $(\eta_t)$
**Initialisation:** Initial point $\hat{\mu}_1 \in \mathcal{K}$, additional information $(\nu_1) \in \mathcal{K}$
**1 For** $t$ in $\{1, \ldots, T-1\}$:
**2**   Update $\hat{\mu}_{temp,t+1} = \hat{\mu}_t - \eta_t \nabla \ell_t(\hat{\mu}_t)$
**3**   Observe $\nu_{t+1}$,
**4**   $\hat{\mu}_{t+1} = f(t, \nu, \hat{\mu}_{temp,t+1})$
**5 Return** $\hat{\mu} = (\hat{\mu}_t)_{t=1,\ldots,T}$

---

**Recovering classical algorithms in the GB-OL framework.** The role of the judge $f$ is to determine, at time $t$, how to combine the expert $\nu$ with the information provided by the gradient descent $\hat{\mu}_{temp,t+1}$. The choice of the judge depends on the confidence we have in $\nu$. For instance, assume that $\nu_t$ at time $t$ is given by an approximation of $\mu_{t-1}^*$ given by multiple gradient descent steps on $\ell_{t-1}$ (which is assumed accessible at time $t$). In this case both OMGD (Zhang et al., 2017) and OGD can be seen as GB-OL algorithms. Indeed, OMGD is a GB-OL algorithm with judge $f(t, \nu, \hat{\mu}_{temp,t+1}) = \nu_{t+1}$. This corresponds to the case where the judge estimates that the additional knowledge is perfectly relevant for the next prediction. OGD is a GB-OL algorithm with judge $f(t, \nu, \hat{\mu}_{temp,t+1}) = \hat{\mu}_{temp,t+1}$. This corresponds to the case where the judge estimates that the additional knowledge is useless or adversarial and then chooses to ignore it.

Allowing different types of expert advice, OptMD can also be seen as a GB-OL algorithm when the regularisation function is the squared distance. Then $f$ is $f(t, \nu, \hat{\mu}_{temp,t+1}) = \hat{\mu}_{temp,t+1} - \eta \nu_t$, where $\eta$ is the gradient step. This judge is less naive than OMGD and OGD on combining additional knowledge and gradient step while remaining fairly confident in $\nu$ as it uses this additional knowledge in any case.

## 1.2 Optimistically tempered online learning

The general framework of Sec. 1.1 allows characterizing optimistic algorithms as follows: *a GB-OL algorithm is optimistic if expert advice is considered independently of the gradient term and if the judge does not ignore experts*, this covers in particular OptMD and OMGD. However, one may wonder about the case where experts provide information whose quality is uncertain. It is legitimate to exploit them, but only if we can attenuate its impact if we realize that this additional knowledge is not useful. We refer to this setting as *optimistically tempered online learning (OT-OL)*.

In other words, the OT-OL framework aims to provide algorithms with a weaker confidence assumption on the experts than the optimistic framework. In doing so, we aim to derive algorithms where experts can be exploited even if one is unsure of their usefulness. OT-OL already appeared for linear losses in Bhaskara et al. (2020) as a follow-up of Dekel et al. (2017), however, we go a step further by combining tempered optimism with GB-OL algorithms in a general framework.

A way to fit the OT-OL framework would be to consider online model selection (*e.g.* Orabona, 2014; Wintenberger, 2017). Thus, it is possible to attenuate the confidence we have in a single expert relative to others. However, this approach requires at least two experts with non-similar advice to be efficient. Such knowledge is not always available in practice because of prohibitive computational costs.

**Contributions and outline.** In this work, we investigate a different route, we propose novel gradient-based OT-OL algorithms allowing a single expert, not necessarily trusted. Those algorithms come with sound theoretical guarantees in the form of D-Regret bounds, while most of the existing guarantees of online model aggregation (seen as OT-OL procedures) are S-Regret bounds. Thus, our work is in line with Jadbabaie et al. (2015), while going beyond optimism to reach OT-OL.

Our results are based on: *(i)* a novel judge named ADJUST (see Sec. 2) fitting the OT-OL framework which adjusts the candidate predictor (*e.g.* the OGD update) with respect to the expert advice, *(ii)* the procedure CONSTRUCT which generates the expert advice from multiple gradient descent steps on the current loss.

This combination yields *optimistically tempered (OT)* versions of three classical online algorithms: Online Gradient Descent (OGD, Zinkevich, 2003), Online Newton Step (ONS, Hazan et al., 2007) and AdaGrad (Duchi et al., 2011). Those optimistically tempered versions allow to adapt S-Regret proofs of Hazan (2019) to D-Regret proofs. This leads to D-Regret worst-case guarantees that hold for strongly convex losses: in particular the losses are not necessarily smooth. This focus on non-smooth losses is novel in the dynamic regret field and has been recently studied by Baby and Wang (2022). Note that our guarantees hold for any expert advice satisfying technical conditions (notably satisfied by CONSTRUCT but going beyond it).

More precisely, we present fully empirical D-Regret bounds for expert advice $\nu$ (detailed in Sec. 3) which depend on $P_T(\nu)$, $S_T(\nu)$ instead of $P_T(\mu^*)$, $S_T(\mu^*)$. This is noticeable as we do not need to know the true minimizers to reach an empirical upper bound. Our D-Regret bounds have the following form:

$$\text{D-Regret}_T(\hat{\mu}, \mu^*) \leq f\left(P_T(\nu), S_T(\nu)\right) + g(T).$$

Our main results are gathered in Thms. 3.1, 3.3 and 3.5. A key takeaway message is that we decorrelate the impact of the time horizon $T$ from the impact of the path lengths $P_T$, $S_T$. Our bounds feature a sum of two terms: a function $g(T)$ and a function $f(P_T, S_T)$ combining the different paths. Those results differ from the (optimal) state-of-the-art bound for convex functions of Zhang et al. (2018a, Theorem 4) which is in $\mathcal{O}(\sqrt{T(1 + P_T)})$. Such a decoupling allows us to pin down more precisely what costs in the learning process, be it the optimization phase or the complexity of the problem.

Furthermore, the optimistically tempered versions of OGD, ONS, and AdaGrad provably satisfy D-Regret bounds on the loss sequence defined for any $t \geq 1$: $\mathbb{E}_{t-1}[\ell_t] = \mathbb{E}[\ell_t \mid \mathcal{F}_{t-1}]$ with $(\mathcal{F}_t)_{t \geq 1}$ a filtration adapted to the environment and $\mathbb{E}_{t-1}[\ell_t]$. This ensures that our predictors are robust to the randomness of the environment. Thus, we define the *Dynamic Cumulative Risk* (D-C-Risk) (already introduced in Wintenberger, 2024) as follows: for any predictable[2] sequences $\hat{\mu}$ and $\mu$ of predictors (i.e.,, $\hat{\mu}_t$ and $\mu_t$ are $\mathcal{F}_{t-1}$ measurable), we denote $L_t = \mathbb{E}_{t-1}[\ell_t]$, $t \geq 1$, and D-C-Risk$_T(\hat{\mu}, \mu) = \sum_{t=1}^{T} L_t(\hat{\mu}_t) - \sum_{t=1}^{T} L_t(\mu_t)$, $T \geq 1$.

Our novel algorithms then satisfy dynamic cumulative risks of the following form: for any predictable sequences $\hat{\mu}$ and $\mu$, any expert advice $\nu$, with probability at least $1 - \delta$,

$$\text{D-C-Risk}_T(\hat{\mu}, \mu) \leq f(P_T(\nu), S_T(\nu)) + g(T, \delta).$$

---

[2] in the sense that predictors only depend on the past.

Those results, gathered in Thms. 3.2, 3.4 and 3.6, are universal in the sense that our comparators can be any predictable sequence and pessimistic as the bound does not involve those comparators.

Finally, we perform experiments (Sec. 4) to assess our algorithm's efficiency. In particular, we test one of our methods (OT-OGD) on several real-life datasets to compare its performance to OGD or OMGD. The comparison with OMGD is particularly relevant since our theoretical results, while slightly weaker than Zhang et al. (2017); Zhao and Zhang (2021, Corollary 4), have a broader range of application, and require weaker assumptions than those Zhang et al. (2017) and different than Zhao and Zhang (2021) (strong convexity vs. convexity and smoothness). We also propose a toy experiment illustrating the interest of using an OT-OL algorithm instead of OGD and OMGD involving dynamic cumulative risks: tempering the impact of expert advice is beneficial for learning. We close this work with some additional technical background (Appendix A), further details on motivation (Appendix B), and we defer to Appendices C and D the proofs of the results of Sec. 3.

## 2 A new optimistically tempered judge

**Framework.** In this work (unless explicitly precised), we use the following mathematical objects and their associated assumptions. First, the set of predictors $\mathcal{K} \subseteq \mathbb{R}^d$ is a closed convex set with finite diameter $D$. Second, we denote by $||.||$ the Euclidean norm on $\mathbb{R}^d$. Also, our loss functions $(\ell_t)_{t \geq 1}$ are $\lambda$-strongly convex:

$$\forall (t, \mu, \mu_0) \in \mathbb{N}/\{0\} \times \mathcal{K}^2, \ell_t(\mu) - \ell_t(\mu_0) \leq \langle \nabla \ell_t(\mu), \mu - \mu_0 \rangle - \lambda \|\mu - \mu_0\|^2.$$

Finally, all gradients are bounded by some constant $G$: $\forall t \geq 1, \mu \in \mathcal{K}, ||\nabla \ell_t(\mu)|| \leq G$.

**The ADJUST algorithm.** We introduce an optimistically tempered judge (namely ADJUST, Algorithm 2) which adjusts a candidate predictor (*e.g.*, obtained through classical OGD) with respect to expert advice. In what follows, we consider those advice as an *additional knowledge* which consists in a sequence of vectors belonging to $\mathbb{R}^d$. In the OT-OL spirit, this knowledge has to be carefully infused into the algorithm. To do so, we exploit the additional knowledge through the notion of *performance*.

**Definition 2.1.** *We use the notation $\langle x, y \rangle_H := x^T H y$ to denote the inner product associated to a positive definite matrix $H$. For a sequence of additional knowledge $\nu = (\nu_t)_{t \geq 0}$, a sequence $\hat{\mu}_{temp} = (\hat{\mu}_{temp,t})_{t \geq 1} \in \mathcal{K}^{\mathbb{N}}$ (the output of a classical online procedure) and for any positive definite matrix $H$, one defines the* performance *at time $t$ of $\hat{\mu}_{temp}$ with regards to $\nu, H$ as follows: we set $m_t := \frac{\nu_{t+1} + \nu_t}{2}$ and*

$$\text{Perf}(t, H, \hat{\mu}_{temp}, \nu) := \langle \hat{\mu}_{temp,t+1} - m_t, \nu_{t+1} - \nu_t \rangle_H.$$

For more details about this notion of performance, we refer to Appendix B.

**Understanding the performance.** At time $t$, the performance exploits the expert $\nu$ through $m_t$ and $\nu_{t+1} - \nu_t$. The first term is new to the best of our knowledge while the second is similar to Rakhlin and Sridharan (2013a). Indeed, the expert advice of Rakhlin and Sridharan (2013a) provides information on the gradient space, and $\nu_{t+1} - \nu_t$ gives similar information. This point is also highlighted in Jadbabaie et al. (2015) as their path $D_T$ focuses on the distance between additional and the gradient of their predictor.

We now state the algorithm ADJUST (Algorithm 2) which takes as input $\hat{\mu}_{temp}$, $\nu$, $H, t$ as defined in definition 2.1 and outputs an updated predictor $\hat{\mu}_{t+1}$. We denote by $\Pi_{\mathcal{K}, \mathcal{H}}$ the projection over the closed convex set $\mathcal{K}$ with respect to the distance induced by $\langle ., . \rangle_H$. We illustrate in Fig. 1 what ADJUST concretely performs when $H = \mathbf{I}_2$ and $\mathcal{K} = \mathbb{R}^2$.

Fig. 1 is crucial to understand why we call ADJUST an optimistically tempered judge. Indeed, the influence of the expert advice is seen as follows: if the dynamic $\nu_{t+1} - \nu_t$ points in the same direction as $\hat{\mu}_{temp,t+1}$ in the referential centered in $m_t$, then ADJUST considers that the expert does not provide an information which is not contained in the gradient (included in $\hat{\mu}_{temp,t+1}$ in a GB-OL algorithm) and choose then ignore it. Otherwise, $\text{Perf}(t, I, \hat{\mu}_{temp}, \nu) < 0$, meaning that the expert can provide information not contained in the gradient, it then adjusts the gradient trajectory *w.r.t.* the dynamic $\nu_{t+1} - \nu_t$ of the expert.

The mathematical translation of this analysis is that ADJUST makes $\hat{\mu}_{t+1}$ closer from $\nu_{t+1}$ than $\nu_t$, which implies less confidence on the expert than directly involving $\nu_{t+1}$. This is further developed in Lemma 2.2.

---

**Algorithm 2:** The ADJUST algorithm

**Parameters :** Time $t$, positive definite $H$, additional knowledge $(\nu_i)_{i=1..t+1}$, candidate $\hat{\mu}_{temp,t+1}$

**1** Set up $m_t = \frac{\nu_{t+1}+\nu_t}{2}$.

**2 If** $\text{Perf}(t, H, \hat{\mu}_{temp}, \nu) < 0$**, then:**

**3**      Set $\hat{\mu}_{t+1} = \underset{\mu \in \mathcal{K}}{\arg\min} \|2m_t - \hat{\mu}_{temp,t+1} - \mu\|_H^2 \ := \Pi_{\mathcal{K},H}(2m_t - \hat{\mu}_{temp,t+1})$

**4 Else:**

**5**      Set $\hat{\mu}_{t+1} = \underset{\mu \in \mathcal{K}}{\arg\min} \|\hat{\mu}_{temp,t+1} - \mu\|_H^2 \ := \Pi_{\mathcal{K},H}(\hat{\mu}_{temp,t+1})$

**6 Return** $\hat{\mu}_{t+1}$

---

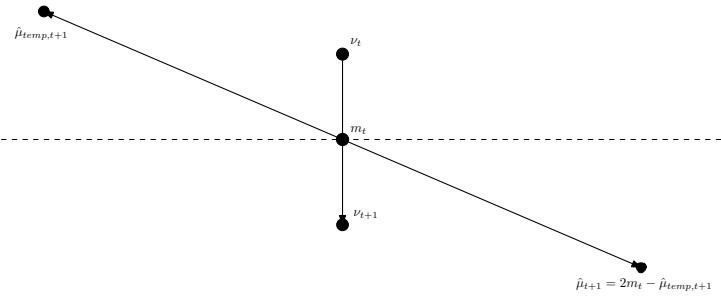

Figure 1: Action of ADJUST when performance is negative

**Lemma 2.2.** *For all $t \geq 0$, any positive definite $H$, any $\hat{\mu}_{temp,t+1}$, $\nu_{t+1}$, $\nu_t$ defined as in ADJUST (algorithm 2): we denote by $\|.\|_H^2$ the norm associated to the scalar product $\langle .,.\rangle_H$.*
*We then have:* $\|\hat{\mu}_{t+1} - \nu_{t+1}\|_H^2 \leq \|\hat{\mu}_{temp,t+1} - \nu_t\|_H^2$.

*Proof of Lemma 2.2.* First, if $\text{Perf}(t+1, H, \hat{\mu}_{temp}, \nu) < 0$, then $\hat{\mu}_{t+1} = \Pi_{\mathcal{K},H}(2m_t - \hat{\mu}_{temp,t+1})$ and one has:

$$\|\hat{\mu}_{t+1} - \nu_{t+1}\|_H^2 = \|\Pi_{\mathcal{K},H}(2m_t - \hat{\mu}_{temp,t+1}) - \nu_{t+1}\|_H^2 \ \leq \|2m_t - \hat{\mu}_{temp,t+1} - \nu_{t+1}\|_H^2$$
$$= \|\hat{\mu}_{temp,t+1} - \nu_t\|_H^2.$$

The last line holding thanks to the definition of $m_t$. Second, if $\text{Perf}(t, H, \hat{\mu}_{temp}, \nu) \geq 0$, we use:

**Lemma 2.3.** *We have* $\forall t \geqslant 0, \|\hat{\mu}_{temp,t+1} - \nu_{t+1}\|_H^2 = \|\hat{\mu}_{temp,t+1} - \nu_t\|_H^2 - 2\text{Perf}(t, H, \hat{\mu}_{temp}, \nu)$.

*Proof of Lemma 2.3.* Recall that $m_t = \frac{\nu_{t+1}+\nu_t}{2}$. We have:

$$\|\hat{\mu}_{temp,t+1} - \nu_{t+1}\|_H^2 = \|\hat{\mu}_{temp,t+1} - m_t + m_t - \nu_{t+1}\|_H^2$$
$$= \|\hat{\mu}_{temp,t+1} - m_t\|_H^2 - \text{Perf}(t, H, \hat{\mu}_{temp}, \nu) + \frac{\|\nu_t - \nu_{t+1}\|_H^2}{4}.$$

And $\|\hat{\mu}_{temp,t+1} - \nu_t\|_H^2 = \|\hat{\mu}_{temp,t+1} - m_t\|_H^2 + \text{Perf}(t, H, \hat{\mu}_{temp}, \nu) + \frac{\|\nu_{t+1} - \nu_t\|_H^2}{4}$.

Thus, $\|\hat{\mu}_{temp,t+1} - \nu_{t+1}\|_H^2 = \|\hat{\mu}_{temp,t+1} - \nu_t\|_H^2 - 2\text{Perf}(t, H, \hat{\mu}_{temp}, \nu)$.      $\square$

Finally, $\|\hat{\mu}_{t+1} - \nu_{t+1}\|_H^2 = \|\Pi_{\mathcal{K},H}(\hat{\mu}_{temp,t+1}) - \nu_{t+1}\|_H^2 \leq \|\hat{\mu}_{temp,t+1} - \nu_{t+1}\|_H^2$
$$= \|\hat{\mu}_{temp,t+1} - \nu_t\|_H^2 - 2\text{Perf}(t, H, \hat{\mu}_{temp}, \nu) \qquad \text{(by Lemma 2.3)}$$
$$\leq \|\hat{\mu}_{temp,t+1} - \nu_t\|_H^2.$$

The last line holding because our performance is positive in this case. This concludes the proof. $\qquad\square$

**Constructing additional knowledge.** We formalize the following data-driven procedure to obtain additional knowledge. We take inspiration from the OMGD algorithm (Zhang et al., 2017). We name this procedure CONSTRUCT and detail it in algorithm 3. It consists of applying $K > 0$ steps of the classical gradient descent algorithm to obtain a good approximation of the last observed minimum.

---

**Algorithm 3:** The CONSTRUCT algorithm.

**Parameters** : The number $K$ of iterations, step-sizes $(\eta_j')_{j=1..K}$
                 Current loss function $\ell_t$, current point $\hat\mu_t$

**Initialisation:** Set $\mathbf{x}_0 := \hat\mu_t$

**1 For** $j$ **in** $0..K-1$**:**

**2** Update
$$\mathbf{x}_{j+1} = \Pi_{\mathcal{K}}\left(\mathbf{x}_j - \eta_j'\nabla\ell_t(\mathbf{x}_j)\right)$$

**3 Return** $\nu_{t+1} := \frac{1}{K}\sum_{j=1}^{K}\mathbf{x}_j$

---

We recall in Lemma 2.4 a convergence property of the gradient descent algorithm.

**Lemma 2.4.** *Assume the considered steps* $(\eta_j')$ *verify for all* $j$, $\frac{1}{\eta_j'} - \lambda \leq \frac{1}{\eta_{j-1}'}$. *Then for any $t$ we have,*

$$\ell_t(\nu_{t+1}) - \ell_t(\mu_t^*) \leq \frac{G^2}{K}\sum_{j=1}^{K}\eta_j'.$$

Proof is deferred to Appendix C. Remark that it is essential to consider strongly convex functions to obtain the rate of Lemma 2.4. To satisfy the technical condition on the step sizes, we can consider the step sequence $(\frac{1}{\lambda t^\alpha})_{t\geq 1}$ for any $\alpha \in [0, 1]$.

## 3 Main results

**Outline.** We present in this section three optimistically tempered variations of OGD, ONS and AdaGrad followed by theoretical guarantees for D-Regret and D-C-Risk. Our theoretical result assumes the CONSTRUCT algorithm but also works for any additional knowledge satisfying technical assumptions (translating here that the experts' advice at time $t$ is a good approximation of the minimum at time $t-1$).

**Proof technique.** Proofs concerning the dynamic regret of our methods (resp. Thms. 3.1, 3.3 and 3.5 ) are gathered in Appendix C and consists in an adaptation of the static proofs of OGD, ONS, and AdaGrad all lying in Hazan (2019). We adapt those proofs using Lemmas 2.2 and 2.4. Proofs concerning the dynamic cumulative risk (resp. Thms. 3.2, 3.4 and 3.6) lie in Appendix D and use the same kind of argument incorporated within the SOCO framework of Wintenberger (2024) described in Appendix D.1.

### 3.1 Optimistically tempered OGD

Our variation of the OGD, called Optimistically Tempered OGD (OT-OGD), is presented in algorithm 4, it exploits an additional information $(\nu_t)_t$ at each time step. Its associated theoretical guarantee for D-Regret is stated in Thm. 3.1.

---

**Algorithm 4:** Projected OT-OGD onto a closed convex space $\mathcal{K}$.

**Parameters** : Horizon $T$, step-sizes $(\eta_t)$

**Initialisation:** Initial point $\mu_1 \in \mathcal{K}$, additional information $(\nu_1) \in \mathcal{K}$

**1 For** $t$ **in** $\{1, \ldots, T\}$**:**

**2**      Update $\hat\mu_{temp,t+1} = \hat\mu_t - \eta_t\nabla\ell_t(\hat\mu_t)$

**3**      Observe $\nu_{t+1}$,

**4**      $\hat\mu_{t+1} = \text{ADJUST}(t, \mathbf{I}_d, (\nu_i)_{i=1..t+1}, \hat\mu_{temp,t+1})$

**5 Return** $\hat\mu = (\hat\mu_t)_{t=0..T}$

---

Note that Algorithm 4 is a GB-OL algorithm with judge ADJUST. This judge aims to be moderated (thus fitting the OT-OL framework) in the sense that it chooses whether the additional knowledge is used given the supplementary information it involves with the information provided by the gradient step.

**Theorem 3.1.** *Denote by $\mu_t^* = \arg\min_{\mu \in \mathcal{K}} \ell_t(\mu)$. We assume that our predictors $\hat{\mu}$ are obtained using OT-OGD (Algorithm 4) with steps $\eta = (\frac{D}{G\sqrt{t}})_{t=1..T}$. We also assume our additional knowledge $\nu$ to be the output of CONSTRUCT (Algorithm 3) used at time $t$ with steps $\eta' = (\frac{1}{\lambda j})_{j=1..K}$ and $K = \lceil \sqrt{T} \rceil$. Then, dynamic regret of OT-OGD with regards to $\mu^* = (\mu_t^*)_{t=0..T}$ the true minimisers satisfy :*

$$\sum_{t=1}^{T} \ell_t(\hat{\mu}_t) - \sum_{t=1}^{T} \ell_t(\mu_t^*) \leq GP_T(\nu) - \lambda S_T(\nu) + \frac{3}{2}GD\sqrt{T} + \frac{G^2}{\lambda}(1 + \log(1+T))\sqrt{T}.$$

*Furthermore, this result remains for any additional knowlege $\nu$ such that for any $t$, $\ell_t(\nu_{t+1}) - \ell_t(\mu_t^*) = \mathcal{O}(\log(t)/\sqrt{t})$.*

Proof is deferred to Appendix C. Thm. 3.1 provides a worst-case guarantee for the dynamic regret of OT-OGD. An interesting point is that our bound decoupled the influence of the path lengths from the horizon $T$, which is not usual in the literature (Zinkevich, 2003 proposed a bound of $\mathcal{O}(\sqrt{T}(1 + P_T))$ later improved in Zhang et al. (2018a) in a $\mathcal{O}(\sqrt{T(1 + P_T)})$).

**Time complexity.** Algorithm 4 can be thought independently of CONSTRUCT when experts are given in advance and satisfy the condition $\ell_t(\nu_{t+1}) - \ell_t(\mu_t^*) = \mathcal{O}(\log(t)/\sqrt{t})$. In this case, OT-OGD has a $\mathcal{O}(T)$ complexity. The use of CONSTRUCT within Algorithm 4 allows to obtain a ready-to-use algorithm, but comes at the cost of an additional time complexity determined by the number of iterations $K$ of Algorithm 3. Here, $K = \lceil \sqrt{T} \rceil$ is similar to the time complexity of the subroutine appearing in the OMGD algorithm of Zhang et al. (2017) with step-size $\eta = 1/\sqrt{T}$. While $K$ depends on the horizon $T$, we can apply CONSTRUCT at each time $t$ with the evolutive number of iterations $K_t = \lceil \sqrt{t} \rceil$. This leads to a D-Regret bound with the same order of magnitude.

**Comparison with literature.** If the true minimiser $\mu_t^*$ is revealed to the learner at time $t + 1$, then taking $\nu_{t+1} = \mu_t^*$ yields $P_T(\nu) = P_{T-1}(\mu^*) + ||\mu_1^* - \nu_1||$. This allows us to compare in this case, our results with those of Zhang et al. (2017). Then, our convergence rate is worse than their $\mathcal{O}(\min(P_T(\mu^*), S_T(\mu^*)))$ while holding with a single strongly convex assumption (no smoothness is required). Our result also holds with different assumptions than the improved rates of Zhao and Zhang (2021) which requires convexity and smoothness. Note however that in the GB-OL framework, this deteriorated rate is not surprising as we pay the shift of an O-OL algorithm (OMGD with judge trusting only the $\nu$ provided by CONSTRUCT, and not the gradient descent step) to an OT-OL algorithm which does not require the optimistic assumption that experts always provide relevant advice. This goes beyond the Optimistic OL framework and highlights the interest of Algorithm 4.

**Role of the path length.** $P_T(\nu)$ Thm. 3.1 does not directly appear in the literature. However in Rakhlin and Sridharan (2013a, Lemma 3), a similar term appears, involving a sum of the distances in the dual space between experts and Nature, Jadbabaie et al. (2015) involves a similar term in the context of dynamic OL. Those terms translate the expert's impact on the training as well as the interplays between experts and the environment. In our study, we decoupled the evolution of $P_T$ and its interplays with the environment. Indeed, in our proofs, we used the following regret decomposition: , if $R = \sum_{t=1}^{T} \ell_t(\hat{\mu}_t) - \sum_{t=1}^{T} \ell_t(\mu_t^*)$, then $R =$

$$\underbrace{\sum_{t=1}^{T} \ell_t(\hat{\mu}_t) - \sum_{t=1}^{T} \ell_t(\nu_t)}_{=(A)} - \underbrace{\sum_{t=1}^{T} \ell_t(\nu_t) - \sum_{t=1}^{T} \ell_t(\nu_{t+1})}_{=(B)} + \underbrace{\sum_{t=1}^{T} \ell_t(\nu_{t+1}) - \sum_{t=1}^{T} \ell_t(\mu_t^*)}_{=(C)}.$$ (A) is dealt via ADJUST and we

choose to separate the terms (B) and (C). Note however, that if we apply directly convexity and bounded gradients on the sum (B)+ (C), we recover a $\mathcal{O}(||\nu_t - \mu_t^*||)$ which captures the environment dynamics. However, assuming directly that $\nu_t$ is closed from $\mu_t^*$ is optimistic, we then relaxed this assumption to $\ell_t(\nu_{t+1}) - \ell_t(\mu_t^*)$ is small (which is more realistic as $\nu_{t+1}$ is $\mathcal{F}_t$ measurable) at the cost of decoupling the evolution of the expert sequence (B) with the performance of $\nu_{t+1}$ wrt the past minimizer.

**Theorem 3.2.** *We assume that our predictors $\hat{\mu}$ are obtained using OT-OGD(Algorithm 4) with steps $\eta = (\frac{D}{G\sqrt{t}})_{t=1..T}$. We also assume our additional knowledge $\nu$ to be the output of CONSTRUCT (Algorithm 3)*

*used at time $t$ with steps $\eta' = (\frac{1}{\lambda j})_{j=1..K}$ and $K = \lceil\sqrt{T}\rceil$. Then, dynamic cumulative risk satisfies with probability $1 - 3\delta$, for any $T \geq 1$, for any sequence $(\mu_t)_{t=1..T}$ such that $\mu_t$ is $\mathcal{F}_{t-1}$-measurable:*

$$\sum_{t=1}^{T} L_t(\hat{\mu}_t) - \sum_{t=1}^{T} L_t(\mu_t) \leq GP_T(\nu) - \lambda S_T(\nu) + \tilde{\mathcal{O}}(\sqrt{T})$$

*where the $\tilde{\mathcal{O}}$ hides a $\log$ factor. Furthermore, this result remains for any additional knowlege $\nu$ such that for any $t$, $\ell_t(\nu_{t+1}) - \ell_t(\mu_t^*) = \mathcal{O}(\log(t)/\sqrt{t})$.*

Proof is deferred to Appendix D. Thm. 3.2 hold for any predictable sequence of comparators $\mu$ which are not involved on the upper bound. This maintains a fully empirical upper bound as predictable sequences are often unknown due to their dependency on the conditional distribution of data. OT-OGD nearly maintains the same convergence rate as Thm. 3.1 while shifting D-Regret for D-C-Regret. As long as paths are sublinear, Thm. 3.2 ensure that the generalization ability of the output of OT-OGD is increasing through time. This is informative on the robustness to the intrinsic randomness of the learning problem. Note that our result holds for any sequence $\mu$ such that $\mu_t$ is $\mathcal{F}_{t-1}$-measurable. We present in Sec. 4.2 a toy experiment that exploits this additional flexibility by showing not only that it may not be relevant to compare ourselves to the true minimizers $\mu^*$, but also that the OT-OL approach outperforms both OGD and OMGD.

### 3.2 Optimistically tempered Online Newton Step

Algorithm 5 details the OT-ONS algorithm, which is an optimistically tempered version of ONS (Hazan et al., 2007). We present in Thm. 3.3 its associated D-Regret bound.

---

**Algorithm 5:** OT-ONS onto a closed convex space $\mathcal{K}$.

**Parameters** : Horizon $T$, step $\gamma, \varepsilon > 0$.
**Initialisation:** convex set $\mathcal{K}$, initial point $\mu_1 \in \mathcal{K} \subseteq \mathbb{R}^d$, additional information $\nu_1 \in \mathcal{K}, A_0 = \varepsilon I_d$
1  **For** $t$ in $\{1, \ldots, T\}$:
2      Update $A_t = A_{t-1} + \nabla_t \nabla_t^\top$
3      Set $\hat{\mu}_{temp,t+1} = \hat{\mu}_t - \frac{1}{\gamma} A_t^{-1} \nabla_t$
4      Observe $\nu_{t+1}$
5      $\hat{\mu}_{t+1} = \text{ADJUST}(t, A_t, \nu, \hat{\mu}_{temp,t+1})$
6  **Return** $\hat{\mu} = (\hat{\mu}_t)_{t=0..T}$

---

**Theorem 3.3.** *Denote by $\mu_t^* = \operatorname{argmin}_{\mu \in \mathcal{K}} \ell_t(\mu)$. We assume that our predictors $\hat{\mu}$ are obtained using OT-ONS (Algorithm 5 ) with $\gamma = \frac{1}{2}\min\left\{\frac{1}{GD}, \alpha\right\}$, $\varepsilon = \frac{1}{\gamma^2 D^2}$. We also assume our additional knowledge $\nu$ to be the output of CONSTRUCT (Algorithm 3) used at time $t$ with steps $\eta' = (\frac{1}{\lambda j})_{j=1..K}$ and $K = T$. Then, dynamic regret of OT-ONS with regards to $\mu^* = (\mu_t^*)_{t=0..T}$ the true minimisers satisfy :*

$$\sum_{t=1}^{T} \ell_t(\hat{\mu}_t) - \sum_{t=1}^{T} \ell_t(\mu_t^*) \leq GP_T(\nu) - \lambda S_T(\nu) + 2\left(\frac{G^2}{\lambda}(d+1) + dGD\right)(1 + \log(T)).$$

*Furthermore, this result remains for any additional knowlege $\nu$ such that for any $t$, $\ell_t(\nu_{t+1}) - \ell_t(\mu_t^*) = \mathcal{O}(1/t)$.*

Thm. 3.3 can be linked to the Online Multiple Newton Update (OMNU) when $\nu_{t+1}$ is the output of multiple Newton steps to approximate $\mu_t^*$, one then can consider OT-ONS as an optimistically tempered version of OMNU. Zhang et al. (2017) proposed a competitive rate of $\mathcal{O}(\min(P_T, S_T))$ for OMNU. While our rate is weaker than theirs, our results hold with the single assumption of strong convexity. Indeed, Zhang et al. (2017, Thm 11.) holds under a set of technical assumptions Zhang et al. (2017, Assumption 10) involving among others, the strict convexity of the losses and holding for problems having small variations of their successive minima. Our result requires fewer assumptions at the cost of $K = T$ iterations of CONSTRUCT at each time step. As OT-ONS is an OT-OL algorithm it is expected to recover a deteriorated rate compared

to OMNU which deals optimistically with experts. Finally, taking $K_t = t$ at each time step allows us to not know in advance the stopping time of OT-ONS and recovers a slightly deteriorated rate of $\mathcal{O}(d\log(T)^2)$.

**Theorem 3.4.** *We assume that our predictors $\hat{\mu}$ are obtained using OT-ONS(Algorithm 5) with $\gamma = \frac{1}{2}\min\left\{\frac{1}{GD}, \frac{\alpha}{4}\right\}$, $\varepsilon = \frac{1}{\gamma^2 D^2}$. We also assume our additional knowledge $\nu$ to be the output of CONSTRUCT (Algorithm 3) used at time $t$ with steps $\eta' = (\frac{1}{\lambda j})_{j=1..K}$ and $K = T$. Then, the dynamic cumulative risk satisfies with probability $1 - 2\delta$, for any $T \geq 1$, for any sequence $(\mu_t)_{t=1..T}$ such that $\mu_t$ is $\mathcal{F}_{t-1}$-measurable:*

$$\sum_{t=1}^{T} L_t(\hat{\mu}_t) - \sum_{t=1}^{T} L_t(\mu_t) \leq GP_T(\nu) + 2G^2 S_T(\nu) + \mathcal{O}(d\log(T) + \log(1/\delta)),$$

*where $L_t = \mathbb{E}_{t-1}[\ell_t]$. This result remains for any additional knowledge $\nu$ s.t. $\forall t, \ell_t(\nu_{t+1}) - \ell_t(\mu_t^*) = \mathcal{O}(1/t)$.*

### 3.3 Optimistically tempered AdaGrad

Algorithm 6 details the OT-Adagrad algorithm, an optimistically tempered version of AdaGrad (Duchi et al., 2011) and we present in Thm. 3.5 its associated D-Regret bound. We use the notation $A \bullet B$ to denote the element-wise multiplication between the matrices $A$ and $B$.

---

**Algorithm 6:** OT-AdaGrad onto a closed convex space $\mathcal{K}$.

**Parameters** : Horizon T, step $\eta$, parameter $\varepsilon$.

**Initialisation:** Initial point $\mu_1 \in \mathcal{K}$, additional information $(\nu_1) \in \mathcal{K}$, $G_0 = \varepsilon\mathbf{I}_d$, $H_0 = G_0^{1/2}$

1 **For** $t$ in $\{1, \ldots, T\}$:
2      Update $G_t = G_{t-1} + \nabla_t \nabla_t^{\top}$
3      Update $H_t = \underset{H \succeq 0}{\arg\min}\left\{G_t \bullet H^{-1} + \text{Tr}(H)\right\} = G_t^{1/2}$
4      Set $\hat{\mu}_{temp,t+1} = \hat{\mu}_t - \eta H_t^{-1}\nabla_t$
5      Observe $\nu_{t+1}$
6      $\hat{\mu}_{t+1} = \text{ADJUST}(t, H_t, \nu, \hat{\mu}_{temp,t+1})$
7 **Return** $\hat{\mu} = (\hat{\mu}_t)_{t=0..T}$

---

**Theorem 3.5.** *Denote by $\mu_t^* = \arg\min_{\mu \in \mathcal{K}} \ell_t(\mu)$. We assume that our predictors $\hat{\mu}$ are obtained using OT-Adagrad (Algorithm 6) with with $\varepsilon = \frac{2}{D^2}, \eta = \frac{D}{\sqrt{2}}$. We also assume our additional knowledge $\nu$ to be the output of CONSTRUCT (Algorithm 3) used at time $t$ with steps $\eta' = (\frac{1}{\lambda j})_{j=1..K}$ and $K = T$. Then, dynamic regret of OT-Adagrad with regards to $\mu^* = (\mu_t^*)_{t=0..T}$ the true minimisers satisfy :*

$$\sum_{t=1}^{T} \ell_t(\hat{\mu}_t) - \sum_{t=1}^{T} \ell_t(\mu_t^*) \leq GP_T(\nu) - \lambda S_T(\nu) + \sqrt{2}D\left(1 + \sqrt{\min_{H \in \mathcal{H}}\sum_t \|\nabla_t\|_H^{*2}}\right) + \frac{G^2}{\lambda}(1 + \log(T)).$$

*Furthermore, this result remains for any additional knowlege $\nu$ such that for any $t$, $\ell_t(\nu_{t+1}) - \ell_t(\mu_t^*) = \mathcal{O}(1/t)$.*

Thm. 3.5 nearly recovers the convergence rate of AdaGrad for static regret at the cost of an extra path length and $\mathcal{O}(\log(T))$ factor. Note that, as in Thm. 3.3, the evolutive iteration number $K_t = t$ can be chosen instead of $K = T$ to make the procedure valid for any horizon $T$ (not necessarily fixed in advance) at the cost of an extra log factor.

Furthermore, Thm. 3.5 goes beyond the scope of Zhang et al. (2017); Zhao and Zhang (2021), as they do not consider AdaGrad. Note that our approach is not the first to propose a dynamic regret bound for AdaGrad (see the recent work of Nazari and Khorram, 2022) however, our approach is, to our knowledge, the first to propose bounds on the D-C-Risk (Thm. 3.6), informing us on the generalization ability of OT-Adagrad.

**Theorem 3.6.** *We assume that our predictors $\hat{\mu}$ are obtained using OT-Adagrad (Algorithm 6) with with $\varepsilon = \frac{2}{D^2}, \eta = \frac{D}{\sqrt{2}}$. We also assume our additional knowledge $\nu$ to be the output of CONSTRUCT (Algorithm 3)*

used at time $t$ with steps $\eta' = (\frac{1}{\lambda j})_{j=1..K}$ and $K = T$. Then, dynamic cumulative risk satisfies with probability $1 - 2\delta$, for any $T \geq 1$, for any sequence $(\mu_t)_{t=1..T}$ such that $\mu_t$ is $\mathcal{F}_{t-1}$-measurable:

$$\sum_{t=1}^{T} L_t(\hat{\mu}_t) - \sum_{t=1}^{T} L_t(\mu_t) \leq GP_T(\nu) + \mathcal{O}\left(\sqrt{\min_{H \in \mathcal{H}} \sum_t \|\nabla_t\|_H^{*2}} + \log\frac{T}{\delta}\right).$$

Note that this result still holds for any additional knowlege $\nu$ such that for any $t$, $\ell_t(\nu_{t+1}) - \ell_t(\mu_t^*) = \mathcal{O}(1/t)$.

## 4 Experiments

This section aims to compare the efficiency of our OT-OL algorithms compared to classical methods. We show here that not being too optimistic *w.r.t.* expert advices leads to comparable or enhanced numerical results. We propose two sets of experiments. The first one gathers 4 classical datasets two regression and two classification problems. Its goal is to assess our algorithm's efficiency by plotting the averaged cumulative losses $\sum_{i=1}^{t} \ell(h_i, z_i)/t$ at any time $t$. The second experiment is a toy example designed to show that D-C-Risk is a relevant tool to handle learning processes on noisy problems. For those two experiments, we compute three algorithms: the celebrated Online Gradient Descent (Zinkevich, 2003, Alg. 1), the OT-OGD algorithm (Algorithm 4) and a variant of the Online Multiple Gradient Descent (OMGD) algorithm with decreasing steps (Zhang et al., 2017, Alg. 1).

The reason we computed OMGD is that CONSTRUCT (Algorithm 3) is following the same idea as OMGD (*i.e.*, performing a gradient descent at each time step for more accurate predictors). An interesting question is whether OT-OGD provides similar or better results than OMGD. We address this below. Furthermore, we would expect that using the output of CONSTRUCT as additional knowledge instead of predictor would provide us additional flexibility in our learning process, is it the case in practice?

### 4.1 Experiments on real-life datasets

We conduct experiments on a few real-life datasets, in classification and regression. Our objective is twofold: check the convergence of our learning methods and compare their efficiencies with classical algorithms.

**Binary Classification.** At each round $t$ the learner receives a data point $x_t \in \mathbb{R}^d$ and predicts its label $y_t \in \{-1, +1\}$ using $\langle x_t, h_t \rangle$, with $h_t$ being the predictor given by the online algorithm of interest. The adversary reveals the true value $y_t$, then the learner suffers the loss $\ell(h_t, z_t) = \left(1 - y_t h_t^T x_t\right)_+$ with $z_t = (x_t, y_t)$ and $a_+ = a$ if $a > 0$ and $a_+ = 0$ otherwise.

**Linear Regression.** At each round $t$, the learner receives a set of features $x_t \in \mathbb{R}^d$ and predicts $y_t \in \mathbb{R}$ using $\langle x_t, h_t \rangle$ with $h_t$ being the predictor given by the online algorithm of interest. Then the adversary reveals the true value $y_t$ and the learner suffers the loss $\ell(h_t, z_t) = \left(y_t - h_t^T x_t\right)^2$ with $z_t = (x_t, y_t)$.

**Datasets.** We consider four real-world datasets: two for classification (Breast Cancer and Pima Indians), and two for regression (Boston Housing and California Housing). All datasets except the Pima Indians have been directly extracted from `sklearn` (Pedregosa et al., 2011). Breast Cancer dataset (Street et al., 1993) is available here and comes from the UCI ML repository as well as the Boston Housing dataset (Belsley et al., 2005) which can be obtained here. California Housing dataset (Pace and Barry, 1997) comes from the StatLib repository and is available here. Finally, Pima Indians dataset (Smith et al., 1988) has been recovered from this Kaggle repository. Note that we randomly permuted the observations to avoid learning irrelevant human ordering of data (such as date or label).

**Parameter settings.** We ran our experiments on a 2021 MacBookPro with an M1 chip and 16 Gb RAM. For OGD, the initialisation point is $\mathbf{0}_{\mathbb{R}^d}$ and the values of the learning rates are set to $\eta = 1/2\sqrt{m}$. where $m$ is the size of the considered dataset. For OMGD, we ran the procedure while, at time $t$, performing a gradient descent with $K = 100$ iterations. This auxiliary gradient descent has been performed with steps $(\lambda/2\sqrt{j})_{j=1..K}$. $\lambda$ ,being an empirical stabiliser set to $0.1/\sqrt{m}$. For OT-OGD, we ran the procedure with a constant step $\eta = 0.1/\sqrt{m}$. We ran CONSTRUCT to generate our additional knowledge with the iteration number $K = 100$ and steps $(\eta'_j)_{j=1..K} = (\lambda/2\sqrt{j})_{j=1..K}$, $\lambda$ ,being an empirical stabiliser set to $0.1/\sqrt{m}$.

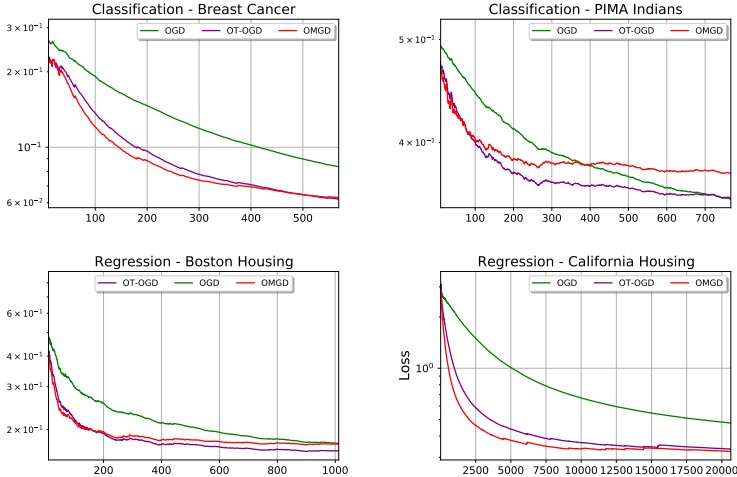

Figure 2: Averaged cumulative losses for all four considered datasets. The $x$-axis is the time. DOGD represents OT-OGD.

**Quantity of interest.** For each dataset, we plot the evolution of the averaged cumulative loss $\sum_{i=1}^{t} \ell\left(h_i, z_i\right)/t$ as a function of the step $t = 1, ..., m$, where $m$ is the dataset size and $h_i$ is the decision made by the learner $h_i$ at step $i$. The results are gathered in Fig. 2.

**Empirical findings.** On those datasets, OMGD with adaptive steps and OT-OGD seem to perform rather equivalently, except on the PIMA Indians dataset where OT-OGD outperforms OMGD. On two datasets (Breast Cancer and California Housing), OT-OGD performs better than OGD, otherwise, both methods perform similarly. A reason that could explain the efficiency of our method compared to OMGD in the PIMA Indians dataset is that because this problem is difficult (*i.e.,* noisy), the technical condition stated in (Zhang et al., 2017, Corollary 4) may not be satisfied. This would impeach OMGD to attain competitive results. Furthermore, note that in any case, OT-OGD is at least as good as OGD or OMGD. The take-home message is that the OT-OL approach is comparably efficient on those datasets.

## 4.2 A toy experiment: the Online Quadratic Problem

**Theoretical framework.** Our problem is set as follows: at each time step $t$, a random variable $\theta_t$ is drawn. For all $t$, $\theta_t$ is such that

$$P_t = \mathcal{L}(\theta_t \mid \mathcal{F}_{t-1}) = \mathcal{N}(\text{moy}_t, \sigma_t^2).$$

We assume that there exists $D_m, D_\sigma$ positive values such that for all $t$, $(\text{moy}_t, \sigma_t) \in [-D_m/2, D_m/2] \times [0; D_\sigma]$. Finally, we consider the losses $\ell_t(\theta) = (\theta_t - \theta)^2$. We refer to this framework as the *Online Quadratic Problem*.

**Quantity of interest.** We study the D-C-Risk w.r.t. the sequence $\mu_t = \text{moy}_t$. We cannot compare ourselves to the true minimizer $\mu_t^* = \theta_t$ because this quantity is not $\mathcal{F}_{t-1}$ measurable. However, we show below that there exists another meaningful comparator. Indeed, in our setup, we note that $\text{moy}_t$ was assumed to be $\mathcal{F}_{t-1}$-measurable so let us see what gives the dynamic cumulative risk for any sequence of predictors $(\hat{\mu}_t)_{t \geq 0}$: $\sum_{t=1}^{T} L_t(\hat{\mu}_t) - \sum_{t=1}^{T} L_t(\text{moy}_t) = \sum_{t=1}^{T} \mathbb{E}_{t-1}[(\theta_t - \hat{\mu}_t)^2] - \sum_{t=1}^{T} \mathbb{E}_{t-1}[(\theta_t - \text{moy}_t)^2] = \sum_{t=1}^{T} (\hat{\mu}_t - \text{moy}_t)^2.$

The last line holding thanks to a bias-variance tradeoff, this basic calculation shows that for this learning problem, using $(\text{moy}_t)_t$ as comparators instead of the true minimizers leads to a meaningful regret. Yet, we can derive from the general notion of dynamic regret a comparison between our prediction and the true mean of the data. One will see in the experiments that OT-OGD can approximate the means better than classical OGD at high times.

**Parameter settings.** All our algorithms are using a projection on the ball centered in 0 of diameter $D = 10$. For OGD, the initialisation point is $\mathbf{0}_{\mathbb{R}^d}$ and the values of the learning rates are set to $\eta_t = 1/2\sqrt{t}$. For OMGD,

we ran the procedure while, at time $t$, performing a gradient descent with $K = 100$ iterations. This auxiliary gradient descent has been performed at time $t$ with steps $(\lambda_t/2\sqrt{j})_{j=1..K}$, $\lambda_t$ being an empirical stabiliser set to $1/2\sqrt{t}$. For OT-OGD, we ran two variants: the first uses CONSTRUCT to generate our additional knowledge. We run algorithm 4 with steps $\eta_t = 1/2\sqrt{t}$ at time $t$. We run CONSTRUCT with, at each time $t$, the iteration number $K = 100$ and steps $(\eta'_j)_{j=1..K} = (\lambda_t/2\sqrt{j})_{j=1..K}$, $\lambda_t$ being an empirical stabiliser set to $1/2\sqrt{t}$. The second does not use CONSTRUCT and instead defines at each time $t$ $\nu_{t+1} \sim \mathcal{N}(\hat{\mu}_t, \sigma_1^2)$ with $\sigma_1 = 0.4$. Similarly, we run algorithm 4 with steps $\eta_t = 1/2\sqrt{t}$ at time $t$.

**Experimental framework.** We take for any $t$, $\mathtt{moy}_t = \sin\left(\frac{t}{\omega}\right)$ with $\omega = 200$, yet the means are a deterministic sequence fixed before our study. Then our $\theta_t$ are drawn independently. We also fix for any $t$, $\sigma_t = \sigma = 4$. We chose $K$ (the number of iterations to acquire our additional knowledge) equal to 100. Results are gathered in Fig. 3.

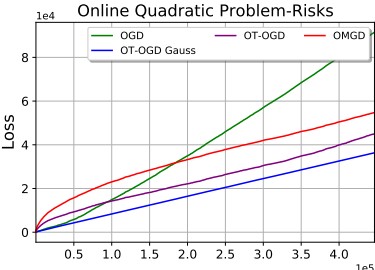 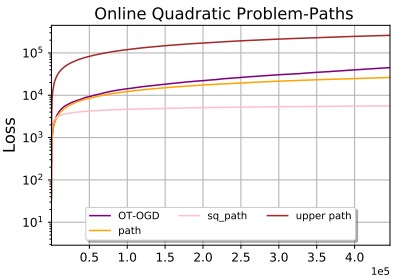

Figure 3: On the left, cumulative risks of OT-OGD (purple,blue), OMGD (red), OGD (green). On the right, plot of OT-OGD and its associated paths. The $x$-axis s the time. $\mathtt{sq\_path}$ is $S_t(\nu)$, $\mathtt{path}$ is $P_t(\nu)$, $\mathtt{upper\_path}$ is $GP_t(\nu) - \lambda S_t(\nu)$.

**Empirical findings.** First, OGD fails on this example as the problem is too noisy: OGD fails to detect any statistical pattern between the successive points. Second, OMGD performs better than OGD but is significantly worse than OT-OGD (the difference of the dynamic cumulative risks is of magnitude $10^4$). This shows that our method, which only uses the output of the auxiliary gradient descent as additional knowledge (and not as predictors as in OMGD) provides flexibility that translates into a greater performance for extremely noisy problems. A reason that could explain the efficiency of our method compared to OMGD is again that the intrinsic noise is so high that the technical condition stated in (Zhang et al., 2017, Cor. 4) may not be satisfied, which impeaches OMGD to attain a competitive dynamic regret in $\mathcal{O}(\min(P_T^*, S_T^*))$. Finally, note that interestingly, our variant of OT-OGD (the curve 'DOGD Gauss' which uses an alternative source of additional information) provides better results here while we have no theoretical guarantee of its efficiency. This opens the way to a broader reflection on the choice of additional knowledge within OT-OGD. In conclusion, this experiment shows that the OT-OL approach outperforms both OGD and OMGD: exhibiting the interest of treating expert advice with caution instead and granting them full trust.

## 5 Conclusion

We introduced the novel Optimistically Tempered Online Learning framework as well as a novel OT judge ADJUST. This judge is flexible enough to provide three optimistically tempered adaptations of classical online methods. To obtain sound D-Regret bounds, we required expert advice to be a good approximation of the local minima. To do so, we exploited CONSTRUCT, however, it is not the only possible choice as we could use, *e.g.*, the Newton algorithm instead. This may be a more suited choice for learning problems in small dimensions as Newton methods are known to converge quickly. This leverages an experimental tradeoff between accuracy and time complexity involving the dimension as a hyperparameter of the problem.

Another promising lead lies in the flexibility of the OT-OL framework when choosing expert advice, as we can confidently propose novel types of advice knowing they will be ignored by ADJUST if useless. More precisely, in this work, $\nu$ focuses on being a good approximation of the minima sequence while our bounds involve a broader tradeoff between path lengths (*i.e.,* only small shifts are recommended through time) and being a good approximation of the past minimizers.

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

## A  Technical background

### A.1  Azuma-Hoeffding's inequality

One recalls the celebrated Azuma- Hoeffding inequality

**Proposition A.1.** *Let $\{X_0, X_1, \cdots\}$ be a martingale with respect to filtration $\{\mathcal{F}_0, \mathcal{F}_1, \cdots\}$. Assume there are predictable processes $\{A_0, A_1, \cdots\}$ and $\{B_0, B_1, \ldots\}$ with respect to $\{\mathcal{F}_0, \mathcal{F}_1, \cdots\}$, i.e. for all $t, A_t, B_t$ are $\mathcal{F}_{t-1}$-measurable, and constants $0 < c_1, c_2, \cdots < \infty$ such that*

$$A_t \leq X_t - X_{t-1} \leq B_t \quad and \quad B_t - A_t \leq c_t$$

*almost surely. Then for all $\epsilon > 0$,*

$$\mathrm{P}\left(|X_n - X_0| \geq \epsilon\right) \leq 2\exp\left(-\frac{2\epsilon^2}{\sum_{t=1}^n c_t^2}\right)$$

In this work we use Azuma-Hoeffding's bound in the particular case where $A_t, B_t$ are constants almost surely.

## B  Inspiration for our notion of performance

Let $\eta = (\eta_t)_{t=1..T)}$ be a positive step sequence.

We denote by $\hat{\mu}_t, t \geq 1$ the sequence of predictors defined by the classical projected OGD:

$$\hat{\mu}_{t+1} = \Pi_{\mathcal{K}}\left(\hat{\mu}_t - \nabla\ell_t(\hat{\mu}_t)\right)$$

**Theorem B.1.** *Dynamic regret of projected OGD on a closed convex $\mathcal{K}$ for convex losses with steps $\eta = (\eta_t)_{t=1..T)}$ with regards to $\mu = (\mu_t)_{t=0..T} \in \mathcal{K}^T$ satisfies :*

$$\sum_{t=1}^T \ell_t(\hat{\mu}_t) - \sum_{t=1}^T \ell_t(\mu_t) \leq \sum_{t=1}^T \langle\nabla(\ell_t), \hat{\mu}_t - \mu_t\rangle$$

$$\leq \frac{D^2}{2\eta_T} + \frac{G^2}{2}\sum_{t=1}^T \eta_t - \sum_{t=1}^T \frac{\mathrm{Perf}(t, \hat{\mu}, \mu)}{\eta_t}.$$

*Proof.* First, convexity of the losses gives us :

$$\sum_{t=1}^T \ell_t\left(\hat{\mu}_t\right) - \sum_{t=1}^T \ell_t\left(\mu_t\right) \leqslant \sum_{t=1}^T \langle\nabla\ell_t\left(\hat{\mu}_t\right), \hat{\mu}_t - \mu_t\rangle$$

To control the right hand side of this bound we use:

$$\|\hat{\mu}_{t+1} - \mu_t\|^2 \leqslant \|\hat{\mu}_t - \eta_t\nabla\ell_t\left(\hat{\mu}_t\right) - \mu_t\|^2$$
$$= \|\hat{\mu}_t - \mu_t\|^2 - 2\eta_t\langle\nabla\ell_t\left(\hat{\mu}_t\right), \hat{\mu}_t - \mu_t\rangle + \eta_t^2\|\nabla\ell_t\left(\hat{\mu}_t\right)\|^2$$

Hence:

$$\|\hat{\mu}_{t+1} - \mu_{t+1}\|^2 \leqslant \|\hat{\mu}_t - \mu_t\|^2 - 2\eta_t\langle\nabla\ell_t\left(\hat{\mu}_t\right), \hat{\mu}_t - \mu_t\rangle + \eta_t^2 G^2 - 2\mathrm{Perf}\left(t, \hat{\mu}, \mu\right)$$

So:

$$\langle\nabla\ell_t\left(\hat{\mu}_t\right), \hat{\mu}_t - \mu_t\rangle \leqslant \frac{\|\hat{\mu}_t - \mu_t\|^2 - \|\hat{\mu}_{t+1} - \mu_{t+1}\|^2}{2\eta_t} + \frac{\eta_t G^2}{2} - \frac{\mathrm{Perf}(t, \hat{\mu}, \mu)}{\eta_t}$$

Summing on $t$ gives (assuming $1/\eta_0 = 0$):

$$\sum_{t=1}^{T} \langle \nabla \ell_t(\hat{\mu}_t), \hat{\mu}_t - \mu_t \rangle \leq \sum_{t=1}^{T} \|\hat{\mu}_t - \mu_t\|^2 \left( \frac{1}{2\eta_t} - \frac{1}{2\eta_{t-1}} \right) + \frac{G^2}{2} \sum_{t=1}^{T} \eta_t - \sum_{t=1}^{T} \frac{\text{Perf}(t, \hat{\mu}, \mu)}{\eta_t}$$

$$\leq D^2 \sum_{t=1}^{T} \left( \frac{1}{2\eta_t} - \frac{1}{2\eta_{t-1}} \right) + \frac{G^2}{2} \sum_{t=1}^{T} \eta_t - \sum_{t=1}^{T} \frac{\text{Perf}(t, \hat{\mu}, \mu)}{\eta_t}$$

$$\leq \frac{D^2}{\eta_T} + \frac{G^2}{2} \sum_{t=1}^{T} \eta_t - \sum_{t=1}^{T} \frac{\text{Perf}(t, \hat{\mu}, \mu)}{\eta_t}$$

$\square$

One can also have a stronger result for $\lambda$-strongly convex functions with the following additional assumption: We assume that our steps $\eta_t$ are such that:

$$\frac{1}{\eta_t} - \lambda \leq \frac{1}{\eta_{t-1}}$$

**Theorem B.2.** *Dynamic regret of projected OGD on a closed convex $\mathcal{K}$ with steps $\eta = (\eta_t)_{t=1..T}$ with regards to $\mu = (\mu_t)_{t=0..T} \in \mathcal{K}^T$ satisfies :*

$$\sum_{t=1}^{T} \ell_t(\hat{\mu}_t) - \sum_{t=1}^{T} \ell_t(\mu_t) \leq \frac{G^2}{2} \sum_{t=1}^{T} \eta_t - \sum_{t=1}^{T} \frac{\text{Perf}(t, \hat{\mu}, \mu)}{\eta_t}.$$

*Proof.* The proof is roughly the same than the one for the previous bound. We remark that thanks to strong convexity, one now has :

$$\sum_{t=1}^{T} \ell_t(\hat{\mu}_t) - \sum_{t=1}^{T} \ell_t(\mu_t) \leqslant \sum_{t=1}^{T} \langle \nabla \ell_t(\hat{\mu}_t), \hat{\mu}_t - \mu_t \rangle - \lambda \|\hat{\mu}_t - \mu_t\|^2$$

So the arguments of the previous proof provide us:

$$\sum_{t=1}^{T} \ell_t(\hat{\mu}_t) - \sum_{t=1}^{T} \ell_t(\mu_t) \leq \frac{1}{2} \sum_{t=1}^{T} \left( \frac{1}{\eta_t} - \lambda \right) \|\hat{\mu}_t - \mu_t\|^2 - \frac{\|\hat{\mu}_{t+1} - \hat{\mu}_{t+1}\|^2}{\eta_t}$$

$$+ \sum_{t=1}^{T} \frac{\eta_t \|\nabla \ell_t(\hat{\mu}_t)\|^2}{2} - \frac{\text{Perf}(t, \hat{\mu}, \mu)}{\eta_t}$$

$$\leq \frac{1}{2} \sum_{t=1}^{T} \frac{\|\hat{\mu}_t - \mu_t\|^2}{\eta_{t-1}} - \frac{\|\hat{\mu}_{t+1} - \hat{\mu}_{t+1}\|^2}{\eta_t}$$

$$+ \sum_{t=1}^{T} \frac{\eta_t \|\nabla \ell_t(\hat{\mu}_t)\|^2}{2} - \frac{\text{Perf}(t, \hat{\mu}, \mu)}{\eta_t}$$

A telescopic argument and bound over the gradients provides us the final result.

$\square$

**Remark B.3.** *We focus in three specific cases where performance can be linked to classical quantities:*

- *First is just a remark : we totally recover the classical OGD bound for static regret when one has $\mu_{t+1} = \mu_t$ for any $t$.*

- *Second, if our OGD predicts well the minimiser $\mu^*$ after a certain time, i.e. for $t \geq t_0$, $\hat{\mu}_{t+1} \approx \mu^*_{t+1}$. Then one has*

$$\sum_{t=1}^{T} \text{Perf}(t, \hat{\mu}, \mu) \approx -\frac{1}{2} \sum_{t=1}^{T} \frac{||\mu^*_{t+1} - \mu^*_t||^2}{\eta_t} \leq -\frac{1}{\eta_1} S^*_T.$$

  *so our result ensures that in this case, OGD has been able to tame the geometry induced by the $\ell_t s$ to generate a momentum greater than $S^*_T/\eta_1$*

- *Finally let us consider the overfitting case i.e, for each $t$, $\hat{\mu}_{t+1} \approx \mu^*_t$. Then:*

$$\sum_{t=1}^{T} \text{Perf}(t, \hat{\mu}, \mu) \approx -\frac{1}{2} \sum_{t=1}^{T} \frac{||\mu^*_{t+1} - \mu^*_t||^2}{\eta_t} \leq \frac{1}{\eta_T} S^*_T.$$

  *So overfitting will penalise our OGD with at most a factor $S_T/\eta_T$*

However, even if our bounds gives us an intuition on how is the OGD interacting with its environment. One cannot control it directly. If we assume having additional information at each time steps, this notion of performance can help us to enhance OGD.

## C Proofs of deterministic results

In this section we use the shortcut $\nabla_t := \nabla \ell_t(\hat{\mu}_t)$.

### C.1 Proof of Lemma 2.4

*Proof.* Let $t \geq 0$. Recall that $\nu_{t+1}$ is defined as the Polyak averaging $\nu_{t+1} := \frac{1}{K} \sum_{j=1}^{K} \mathbf{x}_j$. First, we remark that by convexity of $\ell_t$:

$$\ell_t(\nu_{t+1}) - \ell_t(\mu^*_t) = \ell_t \left( \frac{1}{K} \sum_{j=1}^{K} \mathbf{x}_j \right) - \ell_t(\mu^*_t) \leq \frac{1}{K} \sum_{j=1}^{K} \ell_t(\mathbf{x}_j) - \ell_t(\mu^*_t).$$

Because CONSTRUCT is a gradient descent with steps $(\eta'_j)_{j=1..K}$ on the $\lambda$-strongly convex function $\ell_t$, one has for any $j$, the classical route of proof for static regret bound for strongly convex functions described in (Hazan, 2019, Theorem 3.3). One then has the following, which concludes the proof:

$$\sum_{j=0}^{K} (\ell_t(\mathbf{x}_j) - \ell_t(\mu^*_t)) \leq G^2 \sum_{j=1}^{K} \eta'_j.$$

$\square$

### C.2 A general route of proof

We exhibit in Eq. (1) a general pattern of proof we use several times in this work to bound the dynamic regret. This pattern also structures this document.

$$\sum_{t=1}^{T} \ell_t(\hat{\mu}_t) - \sum_{t=1}^{T} \ell_t(\mu^*_t) = \underbrace{\sum_{t=1}^{T} \ell_t(\hat{\mu}_t) - \sum_{t=1}^{T} \ell_t(\nu_t)}_{=(A)} + \underbrace{\sum_{t=1}^{T} \ell_t(\nu_t) - \sum_{t=1}^{T} \ell_t(\nu_{t+1})}_{=(B)} + \underbrace{\sum_{t=1}^{T} \ell_t(\nu_{t+1}) - \sum_{t=1}^{T} \ell_t(\mu^*_t)}_{=(C)}. \quad (1)$$

Those terms are dealt as follows.

- (A) is controlled by the effect of ADJUST on OGD,ONS,Adagrad. It allows to transform the static guarantees of those algorithms (as stated in Hazan, 2019) into dynamic ones.

- (B) is controlled by the convexity assumptions made on the $\ell_t$s and involve terms like $P_T, S_T$.

- (C) is handled by the way we designed $\nu$.

Our proofs in the rest of this section are based on this general scheme.

Note that we used the sequence $\mu^* = (\mu_t^*)_{t \geq 1}$ as comparators here in order to control (C) via the CONSTRUCT algorithm. This has two implications: *(i)* our results then holds when using any comparator sequence as we control the worst case dynamic regret and *(ii)* we can also involve directly any other comparator sequence $\mu$ within the proof, at the cost of letting (C) unconstrained. We would then need an algorithm different from CONSTRUCT in order to make $\nu_{t+1}$ close to $\mu_t$.

### C.3 Proof of Thm 3.1

**Proposition C.1.** *The sequence of predictors $(\hat{\mu}_t)_{t \geq 0}$ obtained through DOGD on a closed convex $\mathcal{K}$ with steps $\eta = (\eta_t)_{t=1..T)}$ with regards to the additional informations $\nu = (\nu_t)_{t=0..T} \in \mathcal{K}^T$ satisfies :*

$$\sum_{t=1}^{T} \ell_t(\hat{\mu}_t) - \sum_{t=1}^{T} \ell_t(\nu_t) \leq \frac{D^2}{2\eta_T} + \frac{G^2}{2} \sum_{t=1}^{T} \eta_t.$$

*Proof.* We fix $t \geq 0$. For the sake of clarity, we rename $\hat{\mu}_{temp} := \hat{\mu}_{temp,t+1} = \hat{\mu}_t - \eta_t \nabla \ell_t(\hat{\mu}_t)$ (where $\hat{\mu}_{temp,t+1}$ is defined in algorithm 4).

Thanks to convexity of the losses, one has:

$$\sum_{t=1}^{T} \ell_t(\hat{\mu}_t) - \sum_{t=1}^{T} \ell_t(\nu_t) \leq \sum_{t=1}^{T} \langle \nabla \ell_t(\hat{\mu}_t), \hat{\mu}_t - \nu_t \rangle.$$

To control this last sum, our intermediary goal is now to control $||\hat{\mu}_{t+1} - \nu_{t+1}||^2$ in function of $||\hat{\mu}_t - \nu_t||^2$. To do so, we first exploit Lemma 2.2 which stipulates that $||\hat{\mu}_{t+1} - \nu_{t+1}||^2 \leq ||\hat{\mu}_{temp} - \nu_t||^2$. Then we control $\langle \nabla \ell_t(\hat{\mu}_t), \hat{\mu}_t - \nu_t \rangle$.

One has:

$$
\begin{aligned}
||\hat{\mu}_{t+1} - \nu_{t+1}||^2 &\leq ||\hat{\mu}_{temp} - \nu_t||^2 \\
&= ||\hat{\mu}_t - \eta_t \nabla \ell_t(\hat{\mu}_t) - \nu_t||^2 \\
&= ||\hat{\mu}_t - \nu_t||^2 - 2\eta_t \langle \nabla \ell_t(\hat{\mu}_t), \hat{\mu}_t - \nu_t \rangle + \eta_t^2 ||\nabla \ell_t(\hat{\mu}_t)||^2
\end{aligned}
$$

Hence:

$$||\hat{\mu}_{t+1} - \nu_{t+1}||^2 \leqslant ||\hat{\mu}_t - \nu_t||^2 - 2\eta_t \langle \nabla \ell_t(\hat{\mu}_t), \hat{\mu}_t - \nu_t \rangle + \eta_t^2 G^2.$$

So:

$$\langle \nabla \ell_t(\hat{\mu}_t), \hat{\mu}_t - \nu_t \rangle \leqslant \frac{||\hat{\mu}_t - \nu_t||^2 - ||\hat{\mu}_{t+1} - \nu_{t+1}||^2}{2\eta_t} + \frac{\eta_t G^2}{2}.$$

Summing on $t$, gives

$$\sum_{t=1}^{T} \ell_t(\hat{\mu}_t) - \sum_{t=1}^{T} \ell_t(\nu_t) \le \sum_{t=1}^{T} ||\hat{\mu}_t - \nu_t||^2 \left( \frac{1}{2\eta_t} - \frac{1}{2\eta_{t-1}} \right) + \frac{\eta_t G^2}{2}$$
$$\le \frac{D^2}{2\eta_T} + \frac{G^2}{2} \sum_{t=1}^{T} \eta_t.$$

Hence the final result.

$\square$

Now we are able to prove our result:

**Proof of Thm. 3.1**

*Proof.* We control the terms presented in Eq. (1). proposition C.1 ensures us that:

$$(A) \le \frac{D^2}{2\eta_T} + \frac{G^2}{2} \sum_{t=1}^{T} \eta_t$$
$$\le \frac{3}{2} GD\sqrt{T},$$

The last line holding thanks to the definition of $\eta$ and that $\sum_{t=1}^{T} \frac{1}{\sqrt{t}} \le 2\sqrt{T}$.

We now have to deal with (B) and (C) of Eq. (1).

(B) is handled using the strong convexity of $\ell_t$ for any $t$ :

$$\ell_t(\nu_t) - \ell_t(\nu_{t+1}) \le \nabla\ell_t(\nu_t)^\top (\nu_t - \nu_{t+1}) - \lambda||\nu_{t+1} - \nu_t||^2$$
$$\le ||\nabla\ell_t(\nu_t)||.||\nu_{t+1} - \nu_t|| - \lambda||\nu_{t+1} - \nu_t||^2 \qquad \text{Cauchy-Schwarz}$$
$$\le G||\nu_{t+1} - \nu_t|| - \lambda||\nu_{t+1} - \nu_t||^2.$$

Summing over all $t$ gives us :

$$(B) \le G P_T(\nu) - \lambda S_T(\nu).$$

To deal with (C), we exploit Lemma 2.4. Indeed, our choice of steps ensure us that at each step $j$: $\frac{1}{\eta_j'} - \lambda = \lambda(j-1) = \frac{1}{\eta_{j-1}'}$. We have at each time $t$:

$$\ell_t(\nu_{t+1}) - \ell_t(\mu_t^*) \le \frac{G^2}{K} \sum_{j=1}^{K} \eta_j' = \frac{G^2}{\lambda K} \sum_{j=1}^{K} \frac{1}{j}$$
$$\le \frac{G^2(1 + \log(K))}{\lambda K}.$$

Finally:

$$(C) \le T \frac{G^2(1 + \log(K))}{\lambda K}$$
$$\le \frac{G^2}{\lambda} \sqrt{T}(1 + \log(1 + T))$$

The last line holding because $K = \lceil \sqrt{T} \rceil$.

Combining the bounds of (A),(B),(C) concludes the proof.

$\square$

## C.4  Proof of Thm 3.3

We need first to introduce on exp-concave funtion.

**Definition C.2.** *A function $f : \mathbb{R}^n \to \mathbb{R}$ is $\alpha$ exp-concave over a convex $\mathcal{K}$ if the function $g = \exp(-\alpha f)$ is concave on $\mathcal{K}$.*

One also recalls the following lemma coming from (Hazan, 2019, Lemma 4.3)

**Lemma C.3.** *Let $f : \mathcal{K} \to \mathbb{R}$ be an $\alpha$-exp-concave function, and $D, G$ denote the diameter of $\mathcal{K}$ and a bound on the (sub)gradients of $f$ respectively. The following holds for all $\gamma \leq \frac{1}{2} \min \left\{ \frac{1}{4GD}, \alpha \right\}$ and all $\mathbf{x}, \mathbf{y} \in \mathcal{K}$ :*

$$f(\mathbf{x}) \geq f(\mathbf{y}) + \nabla f(\mathbf{y})^\top (\mathbf{x} - \mathbf{y}) + \frac{\gamma}{2} (\mathbf{x} - \mathbf{y})^\top \nabla f(\mathbf{y}) \nabla f(\mathbf{y})^\top (\mathbf{x} - \mathbf{y}).$$

One now states a key preliminary result of this section (proposition C.4) whoch exploits the exp-concavity property.

**Proposition C.4.** *We assume our loss functions $\ell_t$ to be $\alpha$ exp-concave. Let $\{\hat{\mu}_t\}$ being the output of D-ONS (algorithm 5) with $\gamma = \frac{1}{2} \min \left\{ \frac{1}{GD}, \alpha \right\}$, $\varepsilon = \frac{1}{\gamma^2 D^2}$. We then have, for $T > 4$ and any additional knowledge $\nu$:*

$$\sum_{t=1}^{T} \ell_t(\hat{\mu}_t) - \ell_t(\nu_t) \leq 2 \left( \frac{1}{\alpha} + GD \right) d \log(T).$$

*Proof.* The proof is similar to the one of (Hazan, 2019, Thm 4.5) which holds for static regret. We prove Lemma C.5 which is an adaptation of (Hazan, 2019, Lemma 4.6).

**Lemma C.5.** *Let $\{\hat{\mu}_t\}$ being the output of algorithm 5 with $\gamma = \frac{1}{2} \min \left\{ \frac{1}{GD}, \alpha \right\}$, $\varepsilon = \frac{1}{\gamma^2 D^2}$. We then have, for $T > 4$ and any additional knowledge $\nu$:*

$$\sum_{t=1}^{T} \ell_t(\hat{\mu}_t) - \ell_t(\nu_t) \leq \left( \frac{1}{\alpha} + GD \right) \left( 1 + \sum_{t=1}^{T} \nabla_t A_t^{-1} \nabla_t^\top \right).$$

*Proof.* We fix $t \geq 1$ and we first apply Lemma C.3:

$$\ell_t(\hat{\mu}_t) - \ell_t(\nu_t) \leq \nabla_t^\top (\hat{\mu}_t - \nu_t) - \frac{\gamma}{2} (\hat{\mu}_t - \nu_t)^\top \nabla_t \nabla_t^\top (\hat{\mu}_t - \nu_t)$$

Recalling the definition of $\hat{\mu}_{temp,t+1}$, substracting by $\nu_t$ and multiplying by $A_t$ gives us:

$$\hat{\mu}_{temp,t+1} - \nu_t = \hat{\mu}_t - \nu_t - \frac{1}{\gamma} A_t^{-1} \nabla_t \tag{2}$$

and:

$$A_t \left( \hat{\mu}_{temp,t+1} - \nu_t \right) = A_t \left( \hat{\mu}_t - \nu_t \right) - \frac{1}{\gamma} \nabla_t \tag{3}$$

Multiplying the transpose of Eq. (2) by Eq. (3) gives us:

$$\left(\hat{\mu}_{temp,t+1} - \nu_t\right)^\top A_t \left(\hat{\mu}_{temp,t+1} - \nu_t\right) = \left(\hat{\mu}_t - \nu_t\right)^\top A_t \left(\hat{\mu}_t - \nu_t\right) - \frac{2}{\gamma} \nabla_t^\top \left(\hat{\mu}_t - \nu_t\right) + \frac{1}{\gamma^2} \nabla_t^\top A_t^{-1} \nabla_t. \quad (4)$$

Our goal is to lower bound the term on left hand-side of this equality. To do so, we first remark

$$\left(\hat{\mu}_{temp,t+1} - \nu_t\right)^\top A_t \left(\hat{\mu}_{temp,t+1} - \nu_t\right) \quad = \left\|\hat{\mu}_{temp,t+1} - \nu_t\right\|_{A_t}^2$$

Because $A_t$ is a positive definite matrix, Lemma 2.2 holds, which allows us to say that $\left\|\hat{\mu}_{temp,t+1} - \nu_t\right\|_{A_t}^2 \geq \left\|\hat{\mu}_{t+1} - \nu_{t+1}\right\|_{A_t}^2$. Thus:

$$\left(\hat{\mu}_{temp,t+1} - \nu_t\right)^\top A_t \left(\hat{\mu}_{temp,t+1} - \nu_t\right) \geq \left\|\hat{\mu}_{t+1} - \nu_{t+1}\right\|_{A_t}^2$$
$$= \left(\hat{\mu}_{t+1} - \nu_{t+1}\right)^\top A_t \left(\hat{\mu}_{t+1} - \nu_{t+1}\right)$$

This fact together with Eq. (4) gives:

$$\nabla_t^\top \left(\hat{\mu}_t - \nu_t\right) \leq \frac{1}{2\gamma} \nabla_t^\top A_t^{-1} \nabla_t + \frac{\gamma}{2} \left(\hat{\mu}_t - \nu_t\right)^\top A_t \left(\hat{\mu}_t - \nu_t\right)$$
$$- \frac{\gamma}{2} \left(\hat{\mu}_{t+1} - \nu_{t+1}\right)^\top A_t \left(\hat{\mu}_{t+1} - \nu_{t+1}\right).$$

Now, summing up over $t = 1$ to $T$ we get that

$$\sum_{t=1}^T \nabla_t^\top \left(\hat{\mu}_t - \nu_t\right) \leq \frac{1}{2\gamma} \sum_{t=1}^T \nabla_t^\top A_t^{-1} \nabla_t + \frac{\gamma}{2} \left(\mu_1 - \nu_1\right)^\top A_1 \left(\mu_1 - \nu_1\right)$$
$$+ \frac{\gamma}{2} \sum_{t=2}^T \left(\hat{\mu}_t - \nu_t\right)^\top \left(A_t - A_{t-1}\right) \left(\hat{\mu}_t - \nu_t\right)$$
$$- \frac{\gamma}{2} \left(\hat{\mu}_{T+1} - \nu_{T+1}\right)^\top A_T \left(\hat{\mu}_{T+1} - \nu_{T+1}\right)$$
$$\leq \frac{1}{2\gamma} \sum_{t=1}^T \nabla_t^\top A_t^{-1} \nabla_t + \frac{\gamma}{2} \sum_{t=1}^T \left(\hat{\mu}_t - \nu_t\right)^\top \nabla_t \nabla_t^\top \left(\hat{\mu}_t - \nu_t\right)$$
$$+ \frac{\gamma}{2} \left(\mu_1 - \nu_1\right)^\top \left(A_1 - \nabla_1 \nabla_1^\top\right) \left(\mu_1 - \nu_1\right)$$

In the last inequality we use the fact that $A_t - A_{t-1} = \nabla_t \nabla_t^\top$, and the fact that the matrix $A_T$ is PSD to bound the last term before the inequality by 0. Thus,

$$\sum_{t=1}^T \ell_t(\hat{\mu}_t) - \ell_t(\nu_t) \leq \frac{1}{2\gamma} \sum_{t=1}^T \nabla_t^\top A_t^{-1} \nabla_t + \frac{\gamma}{2} \left(\mu_1 - \nu_1\right)^\top \left(A_1 - \nabla_1 \nabla_1^\top\right) \left(\mu_1 - \nu_1\right)$$

Using that $A_1 - \nabla_1 \nabla_1^\top = \varepsilon I_n$, $\varepsilon = \frac{1}{\gamma^2 D^2}$ and that $\mathcal{K}$ has a finite diameter $D$ gives us :

$$\sum_{t=1}^T \ell_t(\hat{\mu}_t) - \ell_t(\nu_t) \leq \frac{1}{2\gamma} \sum_{t=1}^T \nabla_t^\top A_t^{-1} \nabla_t + \frac{\gamma}{2} D^2 \varepsilon$$
$$\leq \frac{1}{2\gamma} \sum_{t=1}^T \nabla_t^\top A_t^{-1} \nabla_t + \frac{1}{2\gamma}$$

Since $\gamma = \frac{1}{2} \min\left\{\frac{1}{GD}, \alpha\right\}$, we have $\frac{1}{\gamma} \leq 2\left(\frac{1}{\alpha} + GD\right)$. This gives the lemma.

$\square$

The rest of the proof now follows the exact same route than (Hazan, 2019, Thm 4.5).

**Proof of proposition C.4** First we show that the term $\sum_{t=1}^{T} \nabla_t^\top A_t^{-1} \nabla_t$ is upper bounded by a telescoping sum. Notice that

$$\nabla_t^\top A_t^{-1} \nabla_t = A_t^{-1} \bullet \nabla_t \nabla_t^\top = A_t^{-1} \bullet (A_t - A_{t-1})$$

where for matrices $A, B \in \mathbb{R}^{n \times n}$ we denote by $A \bullet B = \sum_{i=1}^{n} \sum_{j=1}^{n} A_{ij} B_{ij} = \text{Tr}\left(AB^\top\right)$, which is equivalent to the inner product of these matrices as vectors in $\mathbb{R}^{n^2}$.

For real numbers $a, b \in \mathbb{R}_+$, the first order Taylor expansion of the logarithm of $b$ at $a$ implies $a^{-1}(a-b) \leq \log \frac{a}{b}$. An analogous fact holds for positive semidefinite matrices, i.e., $A^{-1} \bullet (A - B) \leq \log \frac{|A|}{|B|}$, where $|A|$ denotes the determinant of the matrix $A$ (this is proved in Hazan, 2019, Lemma 4.7). Using this fact we have

$$\sum_{t=1}^{T} \nabla_t^\top A_t^{-1} \nabla_t = \sum_{t=1}^{T} A_t^{-1} \bullet \nabla_t \nabla_t^\top$$
$$= \sum_{t=1}^{T} A_t^{-1} \bullet (A_t - A_{t-1})$$
$$\leq \sum_{t=1}^{T} \log \frac{|A_t|}{|A_{t-1}|} = \log \frac{|A_T|}{|A_0|}$$

Since $A_T = \sum_{t=1}^{T} \nabla_t \nabla_t^\top + \varepsilon I_n$ and $\|\nabla_t\| \leq G$, the largest eigenvalue of $A_T$ is at most $TG^2 + \varepsilon$. Hence the determinant of $A_T$ can be bounded by $|A_T| \leq \left(TG^2 + \varepsilon\right)^d$. Hence recalling that $\varepsilon = \frac{1}{\gamma^2 D^2}$ and $\gamma = \frac{1}{2} \min\left\{\frac{1}{GD}, \alpha\right\}$, for $T > 4$

$$\sum_{t=1}^{T} \nabla_t^\top A_t^{-1} \nabla_t \leq \log \left(\frac{TG^2 + \varepsilon}{\varepsilon}\right)^d \leq d \log \left(TG^2 \gamma^2 D^2 + 1\right) \leq d \log T$$

Plugging into Lemma C.5 we obtain

$$\sum_{t=1}^{T} \ell_t(\hat{\mu}_t) - \ell_t(\nu_t) \leq \left(\frac{1}{\alpha} + GD\right) (d \log T + 1)$$

which implies the theorem for $d > 1, T \geq 4$.

$\square$

We now can prove Thm. 3.3.

**Proof of Thm. 3.3.**

*Proof.* We control the terms presented in Eq. (1). To deal with (A), we exploit proposition C.4 knowing that a $\lambda$-strongly convex function with its gradient bounded by $G$ is $\lambda/G^2$ exp-concave:

$$(A) \leq 2 \left(\frac{G^2}{\lambda} + GD\right) d(1 + \log(T))$$

We now have to deal with (B) and (C) of Eq. (1).

(B) is handled using the strong convexity of $\ell_t$ for any $t$ :

$$\ell_t(\nu_t) - \ell_t(\nu_{t+1}) \leq \nabla \ell_t(\nu_t)^\top (\nu_t - \nu_{t+1}) - \lambda ||\nu_{t+1} - \nu_t||^2$$
$$\leq ||\nabla \ell_t(\nu_t)||.||\nu_{t+1} - \nu_t|| - \lambda ||\nu_{t+1} - \nu_t||^2 \qquad \text{Cauchy-Schwarz}$$
$$\leq G||\nu_{t+1} - \nu_t|| - \lambda ||\nu_{t+1} - \nu_t||^2.$$

Summing over all $t$ gives us :

$$(B) \leq GP_T(\nu) - \lambda S_T(\nu).$$

To deal with (C), we exploit Lemma 2.4. Indeed, our choice of steps ensure us that at each step $j$: $\frac{1}{\eta'_j} - \lambda = \lambda(j-1) = \frac{1}{\eta'_{j-1}}$. We have at each time $t$:

$$\ell_t(\nu_{t+1}) - \ell_t(\mu_t^*) \leq \frac{G^2}{K} \sum_{j=1}^{K} \eta'_j = \frac{G^2}{\lambda K} \sum_{j=1}^{K} \frac{1}{j}$$
$$\leq \frac{G^2(1 + \log(K))}{\lambda K}.$$

Finally:

$$(C) \leq T \frac{G^2(1 + \log(K))}{\lambda K}$$
$$= \frac{G^2}{\lambda}(1 + \log(T))$$

The last line holding because $K = T$.

Combining the bounds on (A),(B),(C) concludes the proof.

$\square$

### C.5   Proof of Thm 3.5

We first start with a key result for our study of dynamic Adagrad.

**Proposition C.6.** *We assume our loss functions $\ell_t$ to be convex. Let $\{\hat{\mu}_t\}$ being the output of D-Adagrad (algorithm 6) with $\varepsilon = \frac{2}{D^2}, \eta = \frac{D}{\sqrt{2}}$. We then have, for any additional knowledge $\nu$:*

$$\sum_{t=1}^{T} \ell_t(\hat{\mu}_t) - \ell_t(\nu) \leq \sqrt{2}D \left(1 + \sqrt{\min_{H \in \mathcal{H}} \sum_t ||\nabla_t||_H^{*2}}\right)$$

*where $\mathcal{H} := \{X \in \mathbb{R}^{n \times n} \mid Tr(X) \leq 1, X \succeq 0\}$ and for a fixed $H$, $||\mu||_H^{*2} = \mu^T H^{-1}\mu$ where $H^{-1}$ refers to the Moore-Penrose pseudoinverse.*

*Proof.* The proof follows the route of (Hazan, 2019, Thm 5.12) for the full-matrix version of Adagrad. As for dynamic ONS, our only work consists in modifying a lemma of Hazan's proof (Hazan, 2019, Lemma 5.13), the rest holding similarly.

For the sake of completeness, we state all the lemma of interest in this proof, most of them are directly extracted from (Hazan, 2019, Sec.5.6). We start with (Hazan, 2019, Lemma 11).

**Lemma C.7.** *For $H_T$ the last output of Adagrad, we have*

$$\sqrt{\min_{H \in \mathcal{H}} \sum_t \|\nabla_t\|_H^{*2}} = \boldsymbol{Tr}(H_T)$$

We present now our lemma of interest (Hazan, 2019, Lemma 5.13)

**Lemma C.8.**

$$\sum_{t=1}^T \ell_t(\hat{\mu}_t) - \ell_t(\nu_t) \le 2D + \frac{\eta}{2}\left(G_T \bullet H_T^{-1} + \text{Tr}(H_T)\right) + \frac{1}{2\eta}\sum_{t=1}^T (\hat{\mu}_t - \hat{\nu}_t)^\top (H_t - H_{t-1})(\hat{\mu}_t - \nu_t).$$

*Proof.* First, recall that $\sum_{t=1}^T \ell_t(\hat{\mu}_t) - \ell_t(\nu_t) \le \sum_{t=1}^T \nabla_t^\top (\hat{\mu}_t - \nu_t)$.

By the definition of $\hat{\mu}_{temp,t+1}$ :

$$\hat{\mu}_{temp,t+1} - \nu_t = \hat{\mu}_t - \nu_t - \eta H_t^{-1}\nabla_t \tag{5}$$

and multipying by $H_t$ gives:

$$H_t(\hat{\mu}_{temp,t+1} - \nu_t) = H_t(\hat{\mu}_t - \nu_t) - \eta\nabla_t. \tag{6}$$

Multiplying the transpose of Eq. (5) by Eq. (6) we get

$$(\hat{\mu}_{temp,t+1} - \nu_t)^\top H_t(\hat{\mu}_{temp,t+1} - \nu_t)$$
$$= (\hat{\mu}_t - \nu_t)^\top H_t(\hat{\mu}_t - \nu_t) - 2\eta\nabla_t^\top(\hat{\mu}_t - \nu_t) + \eta^2\nabla_t^\top H_t^{-1}\nabla_t. \tag{7}$$

Focusing on the left-hand side of the equality, one remarks that:

$$(\hat{\mu}_{temp,t+1} - \nu_t)^\top H_t(\hat{\mu}_{temp,t+1} - \nu_t) = \|\hat{\mu}_{temp,t+1} - \nu_t\|_{H_t}^2$$

Since $H_t$ is a PD matrix, one can apply Lemma 2.2 to obtain that $\|\hat{\mu}_{t+1} - \nu_{t+1}\|_{H_t}^2 \le \|\hat{\mu}_{temp,t+1} - \nu_t\|_{H_t}^2$.
Applying this result gives:

$$(\hat{\mu}_{temp,t+1} - \nu_t)^\top H_t(\hat{\mu}_{temp,t+1} - \nu_t) \ge \|\hat{\mu}_{t+1} - \nu_{t+1}\|_{H_t}^2$$

This fact together with Eq. (7) gives

$$\nabla_t^\top (\hat{\mu}_t - \nu_t) \le \frac{\eta}{2}\nabla_t^\top H_t^{-1}\nabla_t + \frac{1}{2\eta}\left(\|\hat{\mu}_t - \nu_t\|_{H_t}^2 - \|\hat{\mu}_{t+1} - \nu_{t+1}\|_{H_t}^2\right)$$

Now, summing up over $t = 1$ to $T$ we get that

$$\sum_{t=1}^T \nabla_t^\top (\hat{\mu}_t - \nu_t) \le$$

$$\frac{\eta}{2}\sum_{t=1}^T \nabla_t^\top H_t^{-1}\nabla_t + \frac{1}{2\eta}\|\mu_1 - \nu_1\|_{H_0}^2 + \frac{1}{2\eta}\sum_{t=1}^T\left(\|\hat{\mu}_t - \nu_t\|_{H_t}^2 - \|\hat{\mu}_t - \nu_t\|_{H_{t-1}}^2\right) - \frac{1}{2\eta}\|\hat{\mu}_{t+1} - \nu_{t+1}\|_{H_T}^2$$

$$\le \frac{\eta}{2}\sum_{t=1}^T \nabla_t^\top H_t^{-1}\nabla_t + \sqrt{2}D + \frac{1}{2\eta}\sum_{t=1}^T (\hat{\mu}_t - \nu_t)^\top (H_t - H_{t-1})(\hat{\mu}_t - \nu_t).$$

In the last inequality we used the fact that $\varepsilon = \frac{2}{D^2}$ and bounded $\| \mu_1 - \nu_1 \|$ by $D^2$.

We now prove that $\sum_{t=1}^T \nabla_t^\top H_t^{-1} \nabla_t \leq \left( G_T \bullet H_T^{-1} + \mathrm{Tr}(H_T) \right)$. To this end, define the functions

$$\Psi_t(H) = \nabla_t \nabla_t^\top \bullet H^{-1}, \Psi_0(H) = \mathrm{Tr}(H).$$

By definition, $H_t$ is the minimizer of $\sum_{i=0}^t \Psi_i$ over $\mathcal{H}$ which can be related to a FTL strategy. Thus, using (Hazan, 2019, Lemma 5.4), we have that

$$\sum_{t=1}^T \nabla_t^\top H_t^{-1} \nabla_t = \sum_{t=1}^T \Psi_t(H_t)$$
$$\leq \sum_{t=1}^T \Psi_t(H_T) + \Psi_0(H_T) - \Psi_0(H_0)$$
$$= G_T \bullet H_T^{-1} + \mathrm{Tr}(H_T)$$

This concludes the proof $\qquad\qquad\square$

Lemma C.8 gives us two terms to be bounded. To do so, we use (Hazan, 2019, Lemmas 5.14,5.15) to conclude the proof. Those lemmas are gathered below.

**Lemma C.9.** *For algorithm 6, the following holds*

$$G_T \bullet H_T^{-1} \leq \mathrm{Tr}(H_T).$$

**Lemma C.10.** *Recall that $D$ the Euclidean diameter of $\mathcal{K}$. Then the following bound holds,* $\sum_{t=1}^T \| \mathbf{x}_t - \mathbf{x}^\star \|_{H_t - H_{t-1}}^2 \leq D^2 \mathrm{Tr}(H_T).$

Now combining Lemma C.8 with the above two lemmas, and using $\eta = \frac{D}{\sqrt{2}}$ appropriately, we obtain the theorem.

$\qquad\qquad\square$

We now can prove Thm. 3.5.

**Proof of Thm. 3.5.**

*Proof.* We control the terms presented in Eq. (1). To deal with (A), we exploit proposition C.6:

$$(A) \leq \sqrt{2}D \left( 1 + \sqrt{\min_{H \in \mathcal{H}} \sum_t \| \nabla_t \|_H^{*2}} \right)$$

We now have to deal with (B) and (C) of Eq. (1).

(B) is handled using the strong convexity of $\ell_t$ for any $t$ :

$$\ell_t(\nu_t) - \ell_t(\nu_{t+1}) \leq \nabla \ell_t(\nu_t)^\top (\nu_t - \nu_{t+1}) - \lambda \| \nu_{t+1} - \nu_t \|^2$$
$$\leq \| \nabla \ell_t(\nu_t) \| . \| \nu_{t+1} - \nu_t \| - \lambda \| \nu_{t+1} - \nu_t \|^2 \qquad \text{Cauchy-Schwarz}$$
$$\leq G \| \nu_{t+1} - \nu_t \| - \lambda \| \nu_{t+1} - \nu_t \|^2.$$

Summing over all $t$ gives us :

$$(B) \leq GP_T(\nu) - \lambda S_T(\nu).$$

To deal with (C), we exploit Lemma 2.4. Indeed, our choice of steps ensure us that at each step $j$: $\frac{1}{\eta'_j} - \lambda = \lambda(j-1) = \frac{1}{\eta'_{j-1}}$. We have at each time $t$:

$$\ell_t(\nu_{t+1}) - \ell_t(\mu_t^*) \leq \frac{G^2}{K} \sum_{j=1}^{K} \eta'_j = \frac{G^2}{\lambda K} \sum_{j=1}^{K} \frac{1}{j}$$
$$\leq \frac{G^2(1 + \log(K))}{\lambda K}.$$

Finally:

$$(C) \leq T \frac{G^2(1 + \log(K))}{\lambda K}$$
$$= \frac{G^2}{\lambda}(1 + \log(T))$$

The last line holding because $K = T$.

Combining the bounds on (A),(B),(C) concludes the proof.

$\square$

# D  Proofs of probabilistic results

## D.1  The SOCO framework

In what follows, for a certain filtration $(\mathcal{F}_t)_{t \geq 1}$, we denote by $\mathbb{E}_{t-1}[.] := \mathbb{E}[. \mid \mathcal{F}_{t-1}]$. SOCO's framework has been introduced in Wintenberger (2024). It focuses on a more general notion of regret presented below.

**Definition D.1.** *For loss function $\ell_t$, we denote by $(\mathcal{F}_t)_t$ a filtration s.t. $\ell_t$ is $\mathcal{F}_t$-measurable. For some predictors $(\hat{\mu}_t)_{t=1..T} \in \mathcal{K}$ we define the* dynamic averaged regret *with regards to $(\mu_t)_{t=1..T} \in \mathcal{K}^T$ as follows:*

$$\textit{D-Av-Regret}_T := \sum_{t=1}^{T} \mathbb{E}_{t-1}[\ell_t(\hat{\mu}_t)] - \sum_{t=1}^{T} \mathbb{E}_{t-1}[\ell_t(\mu_t)].$$

We use SOCO here with the two following assumptions:

**(H1)**  The diameter of $\mathcal{K}$ is $D < \infty$ so that $\|x - y\| \leq D, x, y \in \mathcal{K}$, and the functions $\ell_t$ are continuously differentiable over $\mathcal{K}$ a.s. and the gradients are bounded by $G < \infty : \sup_{x \in \mathcal{K}} \|\nabla \ell_t(x)\| \leq G$ a.s.,$t \geq 1$

**(H2)**  The random loss functions $(\ell_t)$ are stochastically exp-concave i.e. it exists $\alpha > 0$ such that, for any $\mu_1, \mu_2 \in \mathcal{K}$:

$$\mathbb{E}_{t-1}[\ell_t(\mu_2)] \leq \mathbb{E}_{t-1}[\ell_t(\mu_1)] + \mathbb{E}_{t-1}[\nabla \ell_t(\mu_2)^T(\mu_2 - \mu_1)] - \frac{\alpha}{2}\mathbb{E}_{t-1}\left[\left(\nabla \ell_t(\mu_2)^T(\mu_2 - \mu_1)\right)^2\right], \quad x, y \in \mathcal{K}.$$

**Remark D.2.** *A $\lambda$-strongly convex function with its gradients bounded by $G$ in absolute value is $\alpha$ stochastically exp-concave with $\alpha = \lambda/G^2$*

Note that Prop 3 of SOCO is valid for dynamic regret:

**Lemma D.3** ((Wintenberger 2021, Proposition 3))**.** *For any decision sequence* $(\hat{\mu}_t)_t \in \mathcal{K}^T, (\mu_t)_t \in (\mathcal{K}^T)^2$, *under* (**H1**) *and* (**H2**)*, with probability* $1 - \delta$*, it holds for any* $\beta > 0$ *and any* $T \geq 1$

$$\sum_{t=1}^{T} \mathbb{E}_{t-1}[\ell_t(\hat{\mu}_t)] - \sum_{t=1}^{T} \mathbb{E}_{t-1}[\ell_t(\mu_t)] \leq \sum_{t=1}^{T} \nabla\ell_t(\hat{\mu}_t)^T(\hat{\mu}_t - \mu_t)$$
$$+ \frac{\beta}{2}\sum_{t=1}^{T}\left(\nabla\ell_t(\hat{\mu}_t)^T(\hat{\mu}_t - \mu_t)\right)^2 + \frac{2}{\beta}\log\left(\delta^{-1}\right)$$
$$+ \frac{\beta - \alpha}{2}\sum_{t=1}^{T}\mathbb{E}_{t-1}\left[\left(\nabla\ell_t(\hat{\mu}_t)^T(\hat{\mu}_t - \mu_t)\right)^2\right]$$

## D.2 Proof of Thm. 3.2

Our goal is now to combine this property with our dynamic OGD. To do so, we want to control the quadratic terms in Lemma D.3. This is the goal of proposition D.4.

**Proposition D.4.** *For any decision sequence* $(\hat{\mu}_t)_t$, *any sequence* $(\mu_t)_t$ *such that for any* $t; (\hat{\mu}_t, \mu_t)$ *is* $\mathcal{F}_{t-1}$-*measurable, with probability* $1 - 2\delta$*, it holds for any* $T \geq 1$

$$\sum_{t=1}^{T}\mathbb{E}_{t-1}[\ell_t(\hat{\mu}_t)] - \sum_{t=1}^{T}\mathbb{E}_{t-1}[\ell_t(\mu_t)] \leq \sum_{t=1}^{T}\nabla\ell_t(\hat{\mu}_t)^T(\hat{\mu}_t - \mu_t) + \left(2(GD)^2 + 6\frac{G^2}{\lambda}\right)\log\left(\delta^{-1}\right)$$

*Proof.* We define $\alpha = \lambda/G^2$ and $Y_t = \nabla\ell_t(\hat{\mu}_t)^T(\hat{\mu}_t - \mu_t)$ remark that $|Y_t| \leq GD$ a.s, we then exploit a corollary of a Poissonian inequality stated in (Wintenberger, 2024, Eq. (7)). With probability $1 - \delta$ we have:

$$\sum_{t=1}^{T}Y_t^2 \leq 2\sum_{t=1}^{T}\mathbb{E}_{t-1}[Y_t^2] + 2(GD)^2\log(1/\delta)$$

Thus, taking an union bound to make hold this inequality simultaneously with the one of Lemma D.3 and taking $\beta$ such that $3\beta - \alpha = 0$ gives us with probability $1 - 2\delta$:

$$\sum_{t=1}^{T}\mathbb{E}_{t-1}[\ell_t(\hat{\mu}_t)] - \sum_{t=1}^{T}\mathbb{E}_{t-1}[\ell_t(\mu_t)] \leq \sum_{t=1}^{T}\nabla\ell_t(\hat{\mu}_t)^T(\hat{\mu}_t - \mu_t) + \left(2(GD)^2 + 6\frac{G^2}{\lambda}\right)\log\left(\delta^{-1}\right)$$

This concludes the proof. □

We are now able to prove Thm. 3.2:

**Proof of Thm. 3.2.**

*Proof.* We first state that for any $(\hat{\mu}_t, \mu_t)$:

$$\sum_{t=1}^{T} \mathbb{E}_{t-1}[\ell_t(\hat{\mu}_t)] - \sum_{t=1}^{T} \mathbb{E}_{t-1}[\ell_t(\mu_t)] = \sum_{t=1}^{T} \mathbb{E}_{t-1}\left[\ell_t(\hat{\mu}_t) - \ell_t(\mu_t)\right]$$

$$\leq \sum_{t=1}^{T} \mathbb{E}_{t-1}\left[\ell_t(\hat{\mu}_t) - \ell_t(\mu_t^*)\right] \qquad \text{with } \mu_t^* = \operatorname{argmin}_{\mu \in \mathcal{K}} \ell_t(\mu)$$

$$= \underbrace{\sum_{t=1}^{T} \mathbb{E}_{t-1}\left[\ell_t(\hat{\mu}_t) - \ell_t(\nu_t)\right]}_{:=S_1} + \underbrace{\sum_{t=1}^{T} \mathbb{E}_{t-1}\left[\ell_t(\nu_t) - \ell_t(\nu_{t+1})\right]}_{:=S_2}$$

$$+ \underbrace{\sum_{t=1}^{T} \mathbb{E}_{t-1}\left[\ell_t(\nu_{t+1}) - \ell_t(\mu_t^*)\right]}_{:=S_3}$$

The sum $S_1$ is controlled by applying proposition D.4. Then the sum $\sum_{t=1}^{T} \nabla \ell_t(\hat{\mu}_t)^T(\hat{\mu}_t - \nu_t)$ is handled by proposition C.1. We then obtain with our specific choice of steps:

$$S_1 \leq \frac{3}{2} GD\sqrt{T} + \left(2(GD)^2 + 6\frac{G^2}{\lambda}\right) \log\left(\delta^{-1}\right) = O(\sqrt{T}).$$

To control the two last sums, we exploit some arguments provided in Thm. 3.1. More precisely we use the bounds designed to control the sum (B) and (C) in the Thm. 3.1's' proof. We then have for any $t \geq 0$, by strong convexity of the losses:

$$\ell_t(\nu_t) - \ell_t(\nu_{t+1}) \leq G||\nu_{t+1} - \nu_t|| - \lambda||\nu_{t+1} - \nu_t||^2.$$

Also, our choice of steps gives for any $j$: $\frac{1}{\eta_j'} - \lambda = \lambda(j-1) = \frac{1}{\eta_{j-1}'}$. Then, using Lemma 2.4 gives:

$$\ell_t(\nu_{t+1}) - \ell_t(\mu_t^*) \leq \frac{G^2}{K} \sum_{j=1}^{K} \eta_j' = \frac{G^2}{\lambda K} \sum_{j=1}^{K} \frac{1}{j}$$

$$\leq \frac{G^2(1 + \log(K))}{\lambda K}.$$

Then, applying our conditional expectations, recalling that $K = \lceil\sqrt{T}\rceil$ and summing over $t$ gives us.

$$S_2 \leq \sum_{t=1}^{T} \mathbb{E}_{t-1}\left[G||\nu_{t+1} - \nu_t|| - \lambda||\nu_{t+1} - \nu_t||^2\right],$$

$$S_3 \leq \frac{G^2}{\lambda}\sqrt{T}(1 + \log(1 + T)) = \tilde{O}(\sqrt{T}).$$

To conclude the proof, one remarks that if one defines

$$M_T := \sum_{t=1}^{T} \mathbb{E}_{t-1}\left[G||\nu_{t+1} - \nu_t|| - \lambda||\nu_{t+1} - \nu_t||^2\right] - (GP_T(\nu) - \lambda S_T(\nu))$$

Then:

$$S_2 \leq \sum_{t=1}^{T} \mathbb{E}_{t-1} \left[ G\|\nu_{t+1} - \nu_t\| - \lambda\|\nu_{t+1} - \nu_t\|^2 \right] = M_T + GP_T(\nu) - \lambda S_T(\nu)$$

$(M_t)_{t\geq 0}$ is a martingale and furthermore for any $t \geq 0$, $-\lambda D^2 \leq \underbrace{G\|\nu_{t+1} - \nu_t\| - \lambda\|\nu_{t+1} - \nu_t\|^2}_{=M_t - M_{t-1}} \leq GD$.

Thus, applying Azuma-Hoeffding's inequality gives us, with probability $1 - \delta$ that $M_T \leq O(\sqrt{T})$

So with probability $1 - \delta$, one has $S_2 \leq GP_T(\nu) - \lambda S_T(\nu) + O(\sqrt{T})$.

Applying an union bound on the bounds of $S_1, S_2$ and summing the bound of $S_1, S_2, S_3$ concludes the proof.

$\square$

### D.3 Proof of Thm. 3.4

*Proof.* We first state that for any $(\hat{\mu}_t, \mu_t)$:

$$\sum_{t=1}^{T} \mathbb{E}_{t-1}[\ell_t(\hat{\mu}_t)] - \sum_{t=1}^{T} \mathbb{E}_{t-1}[\ell_t(\mu_t)] = \sum_{t=1}^{T} \mathbb{E}_{t-1}\left[\ell_t(\hat{\mu}_t) - \ell_t(\mu_t)\right]$$

$$\leq \sum_{t=1}^{T} \mathbb{E}_{t-1}\left[\ell_t(\hat{\mu}_t) - \ell_t(\mu_t^*)\right] \text{ with } \mu_t^* = \operatorname{argmin}_{\mu \in \mathcal{K}} \ell_t(\mu)$$

$$= \underbrace{\sum_{t=1}^{T} \mathbb{E}_{t-1}\left[\ell_t(\hat{\mu}_t) - \ell_t(\nu_{t+1})\right]}_{:=S_1} + \underbrace{\sum_{t=1}^{T} \mathbb{E}_{t-1}\left[\ell_t(\nu_{t+1}) - \ell_t(\mu_t^*)\right]}_{:=S_2}$$

The sum $S_1$ is controlled by applying Lemma D.3. We then obtain with $Y_t = \langle \nabla_t, \hat{\mu}_t - \nu_{t+1} \rangle$ with probability $1 - \delta$ :

$$S_1 \leq \sum_{t=1}^{T} Y_t + \frac{\beta}{2}\sum_{t=1}^{T} Y_t^2 + \frac{\beta - \alpha}{2}\mathbb{E}_{t-1}[Y_t^2] + \frac{2}{\beta}\log(1/\delta).$$

The first sum is controlled by an intermediary result given in Lemma C.5, the second by Cauchy-Schwarz, we then have:

$$\sum_{t=1}^{T} Y_t = \sum_{t=1}^{T}\langle\hat{\mu}_t - \nabla_t, \hat{\nu}_t\rangle + \langle\nabla_t, \hat{\nu}_t - \nu_{t+1}\rangle$$

$$\leq \frac{1}{2\gamma}\sum_{t=1}^{T} \nabla_t^\top A_t^{-1}\nabla_t + \frac{\gamma}{2}\sum_{t=1}^{T} (\hat{\mu}_t - \nu_t)^\top \nabla_t\nabla_t^\top (\hat{\mu}_t - \nu_t) + \frac{1}{2\gamma} + GP_T(\nu)$$

Recall that, because $\gamma = \frac{1}{2}\min(\frac{1}{GD}, \alpha/4)$, $\frac{1}{\gamma} \leq 2\left(\frac{4}{\alpha} + GD\right)$, one has $\sum_{t=1}^{T} \nabla_t^\top A_t^{-1}\nabla_t \leq 2\left(\frac{8}{\alpha} + GD\right)d\log(T)$. Finally, one has:

$$\sum_{t=1}^{T} Y_t \leq 2\left(1 + \frac{8}{\alpha} + GD\right)d\log(T) + \frac{\alpha}{16}\sum_{t=1}^{T}\left(\nabla_t^\top (\hat{\mu}_t - \nu_t)\right)^2 + GP_T(\nu)$$

Plus, remarking that:

$$\left(\nabla_t^\top \left(\hat{\mu}_t - \nu_t\right)\right)^2 = \left(\nabla_t^\top \left(\hat{\mu}_t - \nu_{t+1}\right) + \nabla_t^\top \left(\nu_{t+1} - \nu_t\right)\right)^2 \leq 2Y_t^2 + 2\left(\nabla_t^\top \left(\nu_{t+1} - \nu_t\right)\right)^2$$
$$\leq 2Y_t^2 + 2G^2 \|\nu_{t+1} - \nu_t\|^2$$

Summing on $t$ and reorganising the previous bounds finally gives:

$$S_1 \leq GP_T(\nu) + 2G^2 S_T(\nu) + \frac{\beta + \alpha/4}{2} \sum_{t=1}^T Y_t^2 + \frac{\beta - \alpha}{2} \mathbb{E}_{t-1}[Y_t^2] + \frac{2}{\beta} \log(1/\delta) + O(d \log(T))$$

Finally, because $|Y_t| \leq GD$ a.s, we exploit a corollary of a Poissonian inequality stated in (Wintenberger, 2024, Eq. (7)). With probability $1 - \delta$ we have:

$$\sum_{t=1}^T Y_t^2 \leq 2 \sum_{t=1}^T \mathbb{E}_{t-1}[Y_t^2] + 2(GD)^2 \log(1/\delta) \tag{8}$$

Thus, taking an union bound and $\beta$ such that $3\beta - \alpha/2 = 0$ gives us with probability $1 - 2\delta$:

$$S_1 \leq O(d \log(T)) + GP_T(\nu) + G^2 S_T(\nu) + \left(\frac{12}{\alpha} + \frac{10\alpha}{24}(GD)^2\right) \log(1/\delta)$$

Finally, to control $S_2$, we reuse the arguments provided in Thm. 3.3. More precisely, we use that the step size of CONSTRUCT allow us to use Lemma 2.4 to claim that for any $t \geq 0$:

$$\ell_t(\nu_{t+1}) - \ell_t(\mu_t^*) \leq \frac{G^2}{K} \sum_{j=1}^K \eta_j'$$
$$\leq \frac{G^2(1 + \log(K))}{\lambda K}$$

Then, because $K = T$, applying our conditional expectations and summing over $t$ gives us.

$$S_2 \leq \frac{G^2}{\lambda}(1 + \log(T)) = O(\log(T)).$$

Summing $S_1$ and $S_2$ concludes the proof.

$\square$

### D.4 Proof of Thm. 3.6

*Proof.* We first state that for any $(\hat{\mu}_t, \mu_t)$:

$$\sum_{t=1}^{T} \mathbb{E}_{t-1}[\ell_t(\hat{\mu}_t)] - \sum_{t=1}^{T} \mathbb{E}_{t-1}[\ell_t(\mu_t)] = \sum_{t=1}^{T} \mathbb{E}_{t-1}\left[\ell_t(\hat{\mu}_t) - \ell_t(\mu_t)\right]$$

$$\leq \sum_{t=1}^{T} \mathbb{E}_{t-1}\left[\ell_t(\hat{\mu}_t) - \ell_t(\mu_t^*)\right] \text{ with } \mu_t^* = \operatorname{argmin}_{\mu \in \mathcal{K}} \ell_t(\mu)$$

$$= \underbrace{\sum_{t=1}^{T} \mathbb{E}_{t-1}\left[\ell_t(\hat{\mu}_t) - \ell_t(\nu_{t+1})\right]}_{:=S_1} + \underbrace{\sum_{t=1}^{T} \mathbb{E}_{t-1}\left[\ell_t(\nu_{t+1}) - \ell_t(\mu_t^*)\right]}_{:=S_2}$$

The sum $S_1$ is controlled by applying Lemma D.3. We then obtain with $Y_t = \langle \nabla_t, \hat{\mu}_t - \nu_{t+1} \rangle$ with probability $1 - \delta$ :

$$S_1 \leq \sum_{t=1}^{T} Y_t + \frac{\beta}{2} \sum_{t=1}^{T} Y_t^2 + \frac{\beta - \alpha}{2} \mathbb{E}_{t-1}[Y_t^2] + \frac{2}{\beta} \log(1/\delta).$$

The first sum is controlled by an intermediary result given in proposition C.6, the second by Cauchy-Schwarz, we then have:

$$\sum_{t=1}^{T} Y_t = \sum_{t=1}^{T} \langle \hat{\mu}_t - \nabla_t, \hat{\nu}_t \rangle + \langle \nabla_t, \hat{\nu}_t - \nu_{t+1} \rangle$$

$$\leq \sqrt{2}D \left(1 + \sqrt{\min_{H \in \mathcal{H}} \sum_t \|\nabla_t\|_H^{*2}}\right) + GP_T(\nu)$$

Reorganising the previous bounds finally gives:

$$S_1 \leq GP_T(\nu) + \frac{\beta}{2} \sum_{t=1}^{T} Y_t^2 + \frac{\beta - \alpha}{2} \mathbb{E}_{t-1}[Y_t^2] + \frac{2}{\beta} \log(1/\delta)$$

Finally, because $|Y_t| \leq GD$ a.s, we exploit a corollary of a Poissonian inequality stated in (Wintenberger, 2024, Eq. (7)). With probability $1 - \delta$ we have:

$$\sum_{t=1}^{T} Y_t^2 \leq 2 \sum_{t=1}^{T} \mathbb{E}_{t-1}[Y_t^2] + 2(GD)^2 \log(1/\delta) \tag{9}$$

Thus, taking an union bound and $\beta$ such that $3\beta - \alpha = 0$ gives us with probability $1 - 2\delta$:

$$S_1 \leq \sqrt{2}D \left(1 + \sqrt{\min_{H \in \mathcal{H}} \sum_t \|\nabla_t\|_H^{*2}}\right) + GP_T(\nu) + \left(\frac{2}{\alpha} + \frac{2\alpha}{3}(GD)^2\right) \log(1/\delta)$$

Finally, to control $S_2$, we reuse the arguments provided in Thm. 3.3. More precisely, we use that the step size of CONSTRUCT allow us to use Lemma 2.4 to claim that for any $t \geq 0$:

$$\ell_t(\nu_{t+1}) - \ell_t(\mu_t^*) \leq \frac{G^2}{K} \sum_{j=1}^{K} \eta_j'$$
$$\leq \frac{G^2(1 + \log(K))}{\lambda K}$$

Then, because $K = T$, applying our conditional expectations and summing over $t$ gives us.

$$S_2 \leq \frac{G^2}{\lambda}(1 + \log(T)) = O(\log(T)).$$

Summing $S_1$ and $S_2$ concludes the proof.

$\square$

