# OpenReview forum: "Optimistically Tempered Online Learning"
_TMLR — Rejected by TMLR_

### Review · Reviewer_ccTy · 2024-11-08

**Summary Of Contributions:**

This paper considers online learning problems with strongly-convex loss functions and focuses on dynamic regret, which is the difference between cumulative losses for the algorithm and for the optimal output sequence.
A main contribution of this paper is to propose a new online learning algorithm that exploits experts' advice, in which the framework of optimistic online learning, e.g., by Rakhlin and Sridharan (2013a;b), is followed.
In addition, an algorithm for constructing experts' advice is presented, which leads to dynamic-regret bounds dependent on the path-length of the experts' advice, under the assumption that loss functions are strongly convex.

**Audience:**

Yes

**Claims And Evidence:**

No

**Requested Changes:**

* Please review the points raised under "Strengths and Weaknesses," and if there are any misunderstandings, I would appreciate it if you could point them out.

* When comparing with previous research results, it would be better to specify which algorithm or theorem within those studies is being compared. For example, simply stating "compared with Zhao and Zhang (2021)" does not clarify which specific result in their paper is being referenced.

* Please specify "x-axis is time" in more detail for the numerical experiments. Clarify whether it represents real time, the number of steps, or another measure, and include the units used, if applicable.

Minor comments:
* Caption of Figure 3: The x-axis s the time <- The x-axis is the time?

* page 10, 11: $0_{R^d}$: This is an unfamiliar notation to me; could you explain what it means?

* page 24: **$Tr$**: Is the italic formatting here a typo?

**Strengths And Weaknesses:**

This paper appears to have several issues concerning its comparison with prior research. Firstly, it references $O(\sqrt{TP_T})$ as an optimal or state-of-the-art bound, claiming multiple times (on p.3 and p.7) that the proposed algorithm outperforms this bound or that this comparison is insightful.
However, this comparison seems inappropriate.
The bound in this prior work applies to a more general problem setting that does not assume strong convexity.
In general, the assumption of strong convexity significantly affects the achievable bounds, making it misleading to place results for different problems on an equal footing.

Additionally, while the paper asserts that Zhao and Zhang (2021) make a stronger assumption by assuming smoothness,
I do not necessarily agree.
In fact, in their Theorem 3, a bound of $O(P_T)$ is achieved even in cases without the smoothness assumption, i.e., when there is no upper bound on the parameter $L$.
This can be confirmed by their proof.
If this understanding is correct, the claimed superiority of the proposed results in terms of regret bounds and the strength of assumptions becomes ambiguous.

Finally, the primary technical novelty of this paper seems to lie in Algorithms 2 and 3.
However, it is unclear whether these enhancements are indispensable for achieving the claimed results or what specific effects they have.
A clearer explanation of these aspects would strengthen the persuasiveness of the paper.

---

> ### Author Response · Authors · 2024-11-19
> **Rebuttal**
>
> We thank you for your valuable review, we answer your concerns below.
>
> **Concerning the comparison with optimal rate for convex losses**
>
> Thank you for pointing this. Note that we mentioned this only to mention the discrepancy between the shape of our bounds and such optimal rates. It is true that such comparison is imprecise in page 3 given rates only depending on path lengths exist for strongly convex losses.
>
> In page 7, we forgot to mention a crucial point to justify this comparison: our proof technique allows us to obtain another version of Theorem 3.1 for convex losses and not strongly convex ones. This comes at the cost of making $K=T$ iterations of Construct in OT-OGD instead of $K=\sqrt{T}$, deteriorating the time complexity of OT-OGD. Then we can use the static regret guarantees of OGD instead of Lemma 2.4 to replace the factor $\frac{G^2}{\lambda}\log(1+T)\sqrt{T}$ by $\frac{3}{2}GD\sqrt{T}$ and we also deteriorate the path $GP_T -\lambda S_T$ to $GP_T$. With this precision, it is fairer to denote the discrepancy between the shape of optimal rates for convex function and our result.
>
> We will carefully adjust the comparisons in page 3 and 7 by explicitly referring to this modified version of Theorem 3.1, thank you again.
>
> **Comparison with Zhao and Zhang (2021), Theorem 3**
>
> Thank you for sharing this, we did not read in detail the proof of their Theorem 3, but it indeed appears that only convexity is necessary to obtain a regret bound in $GP_T$. Note that, according to the modified version of Theorem 3.1 mentioned in the previous paragraph, we are able to make appear a $GP_T$ factor as well, at the sole assumption of convex losses, we will detail this.
>
> **About the relevance of our approach**
>
> We would like to clarify what are the useful contributions of this work for the community. The benefits of our approach are threefold:
>
> - *A novel learning paradigm avoiding a confidence assumption on experts* We challenge the confidence assumption on experts  in optimistic online learning by proposing the new framework of *optimistically tempered* (OT) online learning. This allows to incorporate experts with no certainty on their relevance and let an algorithmic *judge* determine their usefulness. We believe that the introduction of this framework could be an interesting basis for the community as it includes both optimistic and non-optimistic methods as OGD, Optimistic Mirror Descent (see section 1.2).  To our knowledge, we provide the first optimistically tempered algorithms with dynamic regret guarantees for both the original losses, quantifying convergence, and their expected counterpart evaluating the robustness of the predictors to the noise of the learning problems.
> - ⁠*Flexibility* Our algorithm Adjust is flexible enough to transform three classical online learning algorithms into optimistically tempered ones, two with fixed learning rates (OGD,ONS), one with adaptive rates (Adagrad).
> - *Performance with respect to OGD and OMGD* Our experiments in Section 4.1 shows that on a few datasets from the UCI collection, our optimistically tempered approach empirically reach the best-of-both worlds between a non-optimistic method (OGD) and an optimistic one (OMGD). More importantly, we exhibit a toy experiment in Section 4.2 where the optimistically tempered approach outperforms the two other approaches  when the learning problem is noisy. In this case, the OMGD algorithm overfits and leads directly to a linear regret on average. Indeed, recall that for the Online Quadratic Problem, the expected regret of interest is $\sum\_{t=1}^T (\hat{\mu}\_t - \texttt{moy}\_t)^2$.  Assume that OMGD has made enough iteration to assume that $\nu\_t = \mu\_{t-1}^*$ for any $t$ and finally recall that $\mu\_{t-1}^*\sim \mathcal{N}(\texttt{moy}\_{t-1},\sigma^2)$. Then in this case, averaging on the draw of $(\mu\_t^*)_{t=1\cdots T}$ yields an expected regret of $\sigma^2 T +\sum\_{t=1}^T  (\texttt{moy}\_t - \texttt{moy}\_{t-1})^2$ thanks to a bias variance decomposition: OMGD pays a factor $\sigma^2T$ growing quadratically with $\sigma$.
> On the contrary, OT-OGD reduces such overfitting by tempering the impact of the experts. Such statement is corroborated by practical experiments.
>
> To highlight those points, we propose to add a shortened version of this list in section 1.3 on a novel paragraph named 'Originality and impact of this work' .
>
> **Other points of inquiry**
>
> - Concerning the comparison with Zhao and Zhang 2021, we were referring to their Theorem 3, we will adjust this paragraph wrt the discussion above.
> - By 'time' in our graphs, we are referring to the horizon $T$ in our theoretical results. We will correct this.
> - Minor comments: thank you for the typos, $0\_{\mathbb{R}^d}$ denotes the vector of $\mathbb{R}^d$ full of $0$.
>
> Again, we thank you for your insightful review and are happy to answer any supplementary question you might have.

---

### Review · Reviewer_rVg4 · 2024-11-09

**Summary Of Contributions:**

This paper considers dynamic regret algorithms for strongly convex losses. The online learner is assumed to be augmented with a sequence of “advice” vector $\nu_1,\dots,\nu_T$, where $\nu_t$ is provided to the learned before it commits to its prediction vector $\hat \mu_t$. The loss at time $t$ $\ell_t$ is assumed to be $\lambda$ strongly convex and $G$-Lipschitz
The papers focuses on designing an algorithm that performs extremely well if it happens that $\nu_t$ is in some way informative about the future loss $\ell_t$, but should still not perform too terribly if it is not informative.

The main regret bound in theorem 3.1 takes the form
$$
\sum_{t=1}^T \ell_t(\hat \mu_t) - \ell_t(\mu_t^\star) \le P(\nu) - \lambda S(\nu)  + DG\sqrt{T}  + \frac{G^2}{\lambda} \log(T) \sqrt{T}
$$
where $P(\nu) = \sum_{t=1}^T \|\nu_t - \nu_{t+1}\|$ and $S(\nu)=\sum_{t=1}^T \|\nu_t-\nu_{t+1\|^2$.

Another bound is provided for the case that the losses are exp-concave. This bound is instead $O(P(\nu) +S(\nu)  + d\log(T))$

Finally, a regret bound is provided using full-matrix adagrad style analysis that is:
$$
 P(\nu) - \lambda S(\nu)  + D\text{trace}\left(\sqrt{\sum_{t=1}^T \nabla_t\nabla_t^\top}\right)  + \frac{G^2}{\lambda} \log(T)
$$

**Audience:**

No

**Claims And Evidence:**

No

**Requested Changes:**

Please address the issues in the review. They are all critical.

**Strengths And Weaknesses:**

I have several concerns about the main theorems in this paper.

## condition on $\nu_t$

The condition on $\nu_t$ that $\ell_t(\nu_{t+1}) -\ ell_t(\mu^\star_t)\le \tilde O(1/\sqrt{t})$ or even $\tilde O(1/t)$ seems extremely restrictive. Indeed, the entire motivation in the intro appears to be the case in which $\nu_t$ is possibly *not* informative, and yet this condition appears to imply that $\nu_t$ *is* very informative!
This condition is written into the theorem is a rather confusing way as well: after the regret bound it is stated "Furthermore, this result remains for any additional knowledge $\nu$ that satisfies...", rather than establishing this as a *necessary condition* prior to the statement of the regret bound. As I understand it, the condition happens to hold using the specific choice of $\nu_t$ provided by the "construct" algorithm and so if we use some other $\nu_t$ sequence the bound may not hold.

In fact, it seems so restrictive that I think just the naive algorithm that always plays $\hat \mu_t = \nu_t$ will already obtain a better bound than presented here. Using the notation and analysis presented in the paper, if we set $\hat \mu_t =\nu_t$, the regret will be:
$$
\sum_{t=1}^T \ell_t(\nu_t) - \ell_t(\mu^\star_t) =(B) + (C) \le  P(\nu) - \lambda S(\nu)  + \frac{G^2}{\lambda} \log(T) \sqrt{T}
$$
This seems strictly better than Theorem 3.1 (and also better than the adagrad-based method Theorem 3.3 and also the ONS based method in Theorem 3.1).
So, is it really necessary to make the restriction on the $\nu$? Or am I missing something in this analysis?


## Improvement on current literature

Even without this issue, I was not able to understand how this paper improves upon the current literature.
Indeed, the only paragraph that explicitly compares to prior regret bounds admits that the bound is worse. The same paragraph seems to claim that this is an unavoidable penalty. I am not sure about this: nowhere do I see provided an explicit and natural example in which the provided algorithm is actually *significantly better* than prior work. Without such an example, it’s not clear why I should be satisfied with any penalty at all. Can the authors please provide such an example?

## exp-concavity vs strong convexity in ONS
In the ONS section, from looking at the notation in the theorem, I expect you are assuming that the losses are no $\alpha$-exp concave, however, this does not appear to be specified in any assumption. In fact, it also looks like $\lambda$ still appears in the final regret bound, and $\alpha$ does not! So, are we still assuming that the losses are $\lambda$-strongly convex? This does not make sense to me.

## full-matrix adagrad bound is strictly weaker than the scalar bound
The Adagrad bound theorem 3.3 uses the bound $\text{trace}\left(\sqrt{\sum_{t=1}^T \nabla_t\nabla_t^\top}\right)$. This bound appears potentially pleasing due to the variational form provided by the original paper and presented in theorem 3.3. However, this bound is actually bad and should not be presented as a good theoretical result without significant justification. The reason is that $\text{trace}\left(\sqrt{\sum_{t=1}^T \nabla_t\nabla_t^\top}\right)$ is actually *always larger* than $\sqrt{\sum_{t=1}^T\|\nabla_t\|^2}$. This is because by concavity of the square root and linearity of trace:
$$\text{trace}\left(\sqrt{\sum_{t=1}^T \nabla_t\nabla_t^\top}\right) \ge\sqrt{\text{trace}\left(\sum_{t=1}^T \nabla_t\nabla_t^\top\right)}=\sqrt{\sum_{t=1}^T \|\nabla_t\|^2}$$
This latter expression can be achieved via ordinary gradient descent with scalar learning rate combined with the doubling trick or via more advanced analysis using the scalar learning rate $1/\sqrt{\sum_{t=1}^T \|\nabla _t\|^2}$, which I expect would be easily applicable here. This would actually make the analysis easier and the result stronger.

If you want to pursue a no-scalar learning rate with theoretical improvements, the *diagonal* adagrad update does actually improve upon scalar gradient descent in some cases.

---

> ### Author Response · Authors · 2024-11-19
> **Rebuttal pt. 1**
>
> We thank you for your detailed review, we answer your various points of inquiry below.
>
>
> **Condition on $\nu_t$**
>
> The condition $\ell(\nu\_{t+1})- \ell(\mu\_t^*) = \mathcal{O}(1/\sqrt{t})$ is less restrictive than you might think, and taking directly $\hat{\mu}\_t= \nu\_t$ may be a less good idea than what you suggest. Indeed, the condition on $\nu$ only forces the expert to provide relevant knowledge wrt the past minimiser and not the current one, and such knowledge may harm learning for problems with high variance as shown in Section 4.2 for the Online Quadratic Problem. More precisely, let us show here that the expected regret of OMGD for the Online Quadratic Problem is linear.
>
> Recall that for the Online Quadratic Problem, the expected regret of interest is $\sum_{t=1}^T (\hat{\mu}\_t - \texttt{moy}\_t)^2$. Indeed, assume that OMGD has made enough iteration to assume that $\nu\_t = \mu\_{t-1}^*$ (the minimiser of $\ell\_{t-1}$) for any $t$ and finally recall that $\mu\_{t-1}^*\sim \mathcal{N}(\texttt{moy}\_{t-1},\sigma^2)$. Then in this case, averaging over the draw of $(\mu\_t^*)\_{t=1\cdots T}$ yields an expected regret of $\mathbb{E}\left[\sum\_{t=1}^T (\hat{\mu}\_t - \texttt{moy}\_t)^2\right] = \sigma^2 T +\sum\_{t=1}^T  (\texttt{moy}\_t - \texttt{moy}\_{t-1})^2$ thanks to a bias variance decomposition. Then in this case, averaging on the draw of $(\mu\_t^*)_{t=1\cdots T}$ yields an expected regret of $\sigma^2 T +\sum\_{t=1}^T  (\texttt{moy}\_t - \texttt{moy}\_{t-1})^2$ thanks to a bias variance decomposition: OMGD pays a factor $\sigma^2T$ growing quadratically with $\sigma$ and linearly with $T$.
>
> We thank you for noticing this and we will add this in Section 4.2 of our work
>
> This example illustrates that our condition on $\nu$ is not trivial. Note also that we challenged this condition in our conclusion at the end of page 12: 'More precisely, in this work, $\nu$ focuses on being a good approximation of the minima sequence while our bounds involve a broader tradeoff between path lengths (i.e., only small shifts are recommended through time) and being a good approximation of the past minimizers'.
>
> We thank you for this discussion and will add those points on the next version of our work.
>
> **Improvement on current literature**
>
> An explicit example where both OMGD (seen as an optimistic algorithm) and OGD (seen as a non-optimistic one) are outperformed by OT-OGD (an optimistically tempered counterpart) is given in Section 4.2. Such statement is corroborated by our experiments in Section 4.2 and the regret bound of Theorem 3.2 suggest an explanation, let us detail this below.
>
> First, we refer to the previous paragraph to see that, when $\nu\_t = \mu\_{t-1}^*$ OMGD has a linear averaged regret of $\sigma^2 T +\sum\_{t=1}^T  (\texttt{moy}_t - \texttt{moy}\_{t-1})^2$, growing quadratically with $\sigma$.
> On the contrary, OT-OGD reduces such overfitting by tempering the impact of the experts. Such statement is corroborated by practical experiments.
>
> Furthermore, all the experiments of Section 4 shows that on 4 proposed UCI datasets and a toy experiment, OT-OGD either matches or outperforms both OGD and OMGD. Those experimental insights illustrate the relevance of the optimistically tempered paradigm in practice. In particular, our experiments shows that the tightest universal regret bound is not necessarily correlated with the best empirical performance.
>
> We realise that we did not insist enough on this point, and we thank you for raising this.
> To improve clarity, we propose to write the following paragraph in the introduction which sums up the benefits of our approach. (see next comment)

---

> ### Author Response · Authors · 2024-11-19
> **Rebuttal pt. 2**
>
> - ⁠*A novel learning paradigm avoiding a confidence assumption on experts* We challenge the confidence assumption on experts in optimistic online learning by proposing the new framework of *optimistically tempered* (OT) online learning. This allows to incorporate experts with no certainty on their relevance and let an algorithmic *judge* determine their usefulness. We believe that the introduction of this framework could be an interesting basis for the community as it includes both optimistic and non-optimistic methods as OGD, Optimistic Mirror Descent (see section 1.2).  To our knowledge, we provide the first optimistically tempered algorithms with dynamic regret for both the original losses, quantifying convergence, and their expected counterpart which evaluates the robustness of the predictors to the intrinsic noise of learning problems.
> - ⁠*Flexibility* Our algorithm Adjust is flexible enough to transform three classical online learning algorithms into optimistically tempered ones, two with fixed learning rates (OGD,ONS), one with adaptive rates (Adagrad). Also, not only provide do we provide OT algorithms, but also convergence and robustness bounds.
> - ⁠*Empirical performance* Our experiments in Section 4.1 shows that on a few datasets from the UCI collection, our optimistically tempered approach empirically reach the best-of-both worlds between a non-optimistic method (OGD) and an optimistic one (OMGD). More importantly, we exhibit a toy experiment in Section 4.2 where the optimistically tempered approach strongly outperforms both OMGD (seen as an optimistic method) and OGD (a non-optimistic one) when the learning problem is noisy. In this case, OMGD overfits on past data and provably implies a linear regret on average.
>
> **Exp-concave vs strong convexity in ONS**
>
> We apologise for the parameter $\alpha$ which is a typo and should be replaced by $\lambda/G^2$. You are right that for ONS , an exp-concave assumption is enough to obtain a good static regret bound. In our proof, exp-concavity is also enough to deal with the sum $\sum_{t=1}^T \ell\_t(\hat{\mu}\_t) - \ell(\nu\_t)$. However, to deal with the sum $\sum\_{t=1}^T \ell(\nu\_{t+1})- \ell(\mu\_t^*)$ strong convexity is necessary to use Lemma 2.4. We will correct this.
>
> **Full matrix Adagrad bound is strictly weaker than a scalar bound**
>
> We thank you for raising this.
>
> We can easily modify the proof of Theorem 3.5 to have guarantees for diagonal Adagrad: in the proof we have to replace the use of Lemma C.7 with Lemma 11 of (Hazan, 2019) for diagonal matrices, and to replace Lemmas C.9 and C.10 with Lemma 5.14 and 5.15 of (Hazan, 2019) for diagonal AdaGrad. Then the rest of the proof follows the same route. We will add it in the next version, and add the discussion about tightness with respect to performance of OGD with the doubling trick.  However, we respectfully disagree that the bound on Full AdaGrad is 'bad' because it is less tight than OGD. Indeed, the bound of full Adagrad may be sublinear if gradients are vanishing, thus it is a useful safety check which ensure the convergence of the method.
>
> Again, we thank you for your meaningful review and are happy to answer any supplementary question you might have.

---

> > ### Comment · Reviewer_rVg4 · 2024-11-26
> > **reply**
> >
> > **Regarding the $\nu$:** I'm still a bit confused here. Your reply doesn't seem to include any analysis of the actual algorithm, only a description of the failure of just playing $\nu_t$. It's completely possible that the proposed algorithm does indeed improve on using $\hat \mu_t = \nu_t$, but as far as I can tell, the theory does not establish this. Regarding the example you provide, it seems to me that the problem is that $P(\nu)$ is linear, and this is why $\nu_t$ obtains linear regret. However, I don't see how the result of Theorem 3.1 would show sublinear regret in this case where the $\nu$ are "not helpful". Indeed, as I mentioned in my review, it looked to me that regret bound resulting from always playing  $\nu_t$ is *strictly better* than the regret bound of Theorem 3.1. Again, the actual regret of the algorithm may be much lower than the bound of Theorem 3.1 since Theorem 3.1 is only an upper-bound, but my confusion is about the bounds. In my reading of the paper, the primary contributions are mathematical and so this is a critical issue.
> >
> > **Regarding the explicit example:** if I understand properly, the explicit example is the quadratic problem mentioned in your response above. Again, I do not see how the result of Theorem 3.1 shows an improvement here. By an explicit example, I mean a problem (or series of problems) for which if you plug in explicit numbers for all the values in Theorem 3.1, the result will be significantly better than some other benchmark bound.
> >
> > So, again I may be missing something, but as far as I could tell, there is *no* setting of the parameters that satisfies the conditions of Theorem 3.1 for which playing $\nu_t$ has a worse regret bound than the result of Theorem 3.1, including the quadratic example you mention. While your experiments suggest the actual algorithm works well (and this is certainly encouraging) this also suggests that the analysis is very loose and does not explain the experiments.
> >
> > Alternatively, if you could identify an explicit point at which my short analysis in the review which is actually incorrect, then that would also make me feel much better about the results.
> >
> > **Regarding full-matrix adagrad:** the quality you mention about decreasing when gradients go zero is not unique to full-matrix adagrad. As mentioned in my review, running online gradient descent with learning rate $\eta_t = \frac{D}{\sqrt{\sum_{i=1}^t \|\nabla_i\|^2}}$ is well-known to ensure the regret bound $R\le O(D\sqrt{\sum_{t=1}^T \|\nabla_t\|^2})$, which is *always better* than the standard full-matrix bound $D\sqrt{\min_{H} \sum_t \|\nabla_t\|_H^\star}$. you are using and is much more efficient to run.
> >
> > Of course, it is entirely possible that there is some other analysis of the full-matrix algorithm that improves the regret bound, but the standard theorem you seem to be using does not. I'm sorry to be a stickler about this, but this is a common misconception and so it's important not to repeat it.
> >
> > If you want to focus on *experiments*, then it's fine to make liberal use of the full-matrix updates - they do often perform quite well empirically! However, from a theoretical perspective the current result does not show any benefit from them. If you want to go this route, then I still think it is best to either not include the worse theoretical analysis, or be very clear that the analysis does not explain any advantage coming from the full-matrix update. This is important because it lets others know that there is still a mystery to resolve here.

---

> > > ### Author Response · Authors · 2024-12-10
> > > **Thank you for your reply**
> > >
> > > Thank you for your time and your answer.
> > >
> > > **About our theoretical results and the information gathered in our Online Quadratic Problem example**
> > >
> > > The example we provided in our previous answer about the online quadratic problem only shows that the expert sequence will provably fail, while optimistically tempered (OT) methods could work. Then we agree with you that for now, we do not have a theoretical proof of efficiency of OT-OL methods compared to existing ones, only empirical evidences. Filling this gap is a promising future lead and, as you said in your last response, there is still a mystery to resolve here.
> > >
> > > However, to our understanding, you seem to consider that, because our bounds do not explain precisely why OT-OL methods work, then they are not of interest. We would like to challenge this point, as our bounds provide a relevant safety check: we have a convergence guarantee as long as the path grows sublinearly.  As you mentioned for the case of full-matrix OT-Adagrad, those guarantees may be generic and probably do not explain why optimistically tempered methods perform better than existing ones (and we are happy to precise it in the next version in order to provide a more lucid take on our results).
> > >
> > > We believe that such a theoretical contribution is in the scope of TMLR, which, as precised in its homepage, 'emphasizes technical correctness over subjective significance, to ensure that we facilitate scientific discourse on topics that may not yet be accepted in mainstream venues but may be important in the future'. This is in this spirit that we developed the Optimistically Tempered framework and associated algorithms, and we are happy to make clear that our main contributions are the introduction of the OT-OL framework, generic regret bounds acting as a safety check for convergence and numerical experiments confirming the potential of OT methods.
> > >
> > > **Regarding full-matrix Adagrad**
> > >
> > > Thank you for raising this, we will make clear that our guarantees remains generic, that the only information they provide is that the method will converge, and that there is room for future work to obtain guarantees which explain precisely why the method work. We are happy to make this clear for all our theoretical results.
> > >
> > > Again, we would like to thank you for your feedback on our work.

---

### Review · Reviewer_mJZ8 · 2024-11-13

**Summary Of Contributions:**

This paper introduces the Optimistically Tempered (OT) Online Learning framework, which integrates expert advice in online learning, adjusting its influence when reliability is uncertain. That is different from traditional optimistic approaches that fully trust expert recommendations.

The authors extend three algorithms—OGD, ONS, and AdaGrad—to the OT framework, introducing OT-OGD, OT-ONS, and OT-AdaGrad, which incorporate the Adjust algorithm (Algorithm 2) to balance gradient updates with expert input. Key theoretical results include:

Theorem 3.1 (OT-OGD), Theorem 3.3 (OT-ONS), and Theorem 3.5 (OT-AdaGrad) provide dynamic regret bounds that separate time horizon and path dependencies, applicable without requiring smoothness.

Theorems 3.2, 3.4, and 3.6 offer cumulative risk bounds, ensuring robustness to environmental noise and improving generalization over time.

Empirical results show that OT algorithms achieve competitive or superior performance in noisy environments, with examples in Figure 2.

This tempered approach enhances online learning with practical robustness and strong theoretical guarantees.

**Audience:**

Yes

**Claims And Evidence:**

Yes

**Requested Changes:**

Overall the paper is very well written and easy to follow. Here are a couple of minor suggestions which may improve the paper.

In Section 3, consider discussing potential ways to extend the OT framework to handle non-convex losses, or at least clarify where strong convexity is essential to the theoretical guarantees. This addition would help readers understand the framework's limitations and where it could be potentially adapted for broader online learning scenarios. This may aslo lead to future investigations bulding on the results of this paper.

In Section 2, the Adjust algorithm’s decision-making process could be made clearer. Specifically, explain how the performance metric in Algorithm 2 interprets when to accept or ignore expert advice and provide a concise example. This would make the algorithm's selective integration mechanism more intuitive, especially for readers less familiar with advanced regret bounds.

**Strengths And Weaknesses:**

**Strengths:**

Novel Framework: The OT framework offers a tempered approach to incorporating expert advice, making it effective for dynamic and noisy environments where advice may be unreliable.

Theoretical Guarantees: The paper provides comprehensive dynamic regret (Theorems 3.1, 3.3, 3.5) and dynamic cumulative risk bounds (Theorems 3.2, 3.4, 3.6), ensuring robustness without requiring smoothness assumptions.

Empirical Support: Experiments on real-world datasets (Figure 2) show OT algorithms performing competitively or better than optimistic methods.

**Weaknesses:**

Dependence on Strong Convexity: The results depend on strong convexity, which may be seen as a limitation.

Computational Overhead: The OT framework, particularly with the Construct algorithm, introduces additional computational complexity. Discussion on scalability and trade-offs in high-dimensional scenarios would benefit readers interested in practical deployment.

---

> ### Author Response · Authors · 2024-11-19
> **Rebuttal**
>
> We warmly thank you for your positive review and happy to read that you found our work 'well written and easy to follow'. We answer your questions below.
>
>  **"In Section 3..."**
>  Thank you for raising this, to put a broader context to our result we precise several points that we will incorporate in our manuscript.
>
>   - A regret bound is reachable for OT-OGD with convex losses : our proof technique allows us to obtain another version of Theorem 3.1 for convex losses and not strongly convex ones. This comes at the cost of making $K=T$ iterations of Construct in OT-OGD instead of $K=\sqrt{T}$, deteriorating the time complexity of OT-OGD. Then we can use the static regret guarantees of OGD instead of Lemma 2.4 to replace the factor $\frac{G^2}{\lambda}\log(1+T)\sqrt{T}$ by $\frac{3}{2}GD\sqrt{T}$ and we also deteriorate the path $GP_T -\lambda S_T$ to $GP_T$.
>   - The crucial usefulness of strong convexity is to deal with experts: Lemma 2.4 allows controlling the distance of experts wrt past minimisers. Note that we acknowledge this limitation in our conclusion and overcoming it is a promising future lead : 'More precisely, in this work, $\nu$ focuses on being a good approximation of the minima sequence while our bounds involve a broader tradeoff between path lengths (i.e., only small shifts are recommended through time) and being a good approximation of the past minimizers'.
>
> **In Section 2...**
> We provided a simple example of the impact of the judge Adjust in Section 2 via Figure 1. A short summary would be that Adjust consider experts when they are suggesting another direction (quantified by the sign of the performance) than a single gradient step. Then the judge considers that novel information is provided and thus incorporate it in the algorithm. We will add this short summary in Section 2 and recall the discussion at the end of page 4 below.
>
> Fig. 1 is crucial to understand why we call Adjust an optimistically tempered judge. Indeed, the influence of the expert advice is seen as follows: if the dynamic $\nu\_{t+1}-\nu\_t$ points in the same direction as $\hat{\mu}\_{temp,t+1}$ in the referential centred in $m_t$, then Adjust considers that the expert does not provide an information which is not contained in the gradient (included in $\hat{\mu}\_{temp,t+1}$ in a GB-OL algorithm) and choose then ignore
> it. Otherwise, $Perf(t, I, \hat{\mu}\_{temp,t+1}, \nu) < 0$, meaning that the expert can provide information not contained in the gradient, it then adjusts the gradient trajectory w.r.t. the dynamic $\nu\_{t+1}-\nu\_t$ of the expert. The mathematical translation of this analysis is that Adjust makes $\hat{\mu}\_{t+1}$ closer to $\nu_{t+1}$ than $\nu\_t$, which implies less confidence on the expert than directly involving $\nu_{t+1}$.
>
>
> Again, we thank you for your positive review and are happy to answer to any additional question you might have.

---

### Decision · Action_Editor_baCs · 2025-01-27

**Recommendation:** Reject

**Comment:**

I regret that this work cannot be accepted at TMLR. Two reviewers expressed concerns about this work, which I summarize here:
- The authors mentioned in their response to Reviewer ccTy that they can obtain another version of Theorem 3.1 (by using $T$ iterations of Construct) to allow Algorithm 3 to work for general convex functions rather than only strongly convex functions. However, the authors did not provide a proof for how this would work, and so it isn’t possible to verify the correctness of this argument.
- The comparison to Zhao and Zhang seems to have issues, which (if I understand the author’s response to ccTY), the authors now acknowledge
- Another reviewer lamented that, at present, it is unclear if there is a regime where the authors' bounds are better than previous works. Based on my read of the discussion, this concern still holds. The authors mention that it might be that they could show bounds for OT-OL that are better than existing bounds, but at this point in time, they do not have such bounds.

On the positive side, the authors seem to have some promising preliminary experimental results. One path forward for this paper, as suggested by another reviewer, would be to emphasize the value of the authors’ OT-OL framework by doing comprehensive experiments, making this work more of an empirical paper. At present, from the theory side, the bounds will not have enough of an audience from the online learning community, and from the empirical side, the work would be considered too light (it seems like primarily a theory work from its current writing).

I also note that one reviewer was more positive, but it is difficult to leave aside the issues mentioned above.

Going forward, I would encourage the authors to revisit their theory and to see if they can try to meet the bar outlined by Reviewer rVg4. To convince a theory audience, it would help tremendously if you could demonstrate regimes (e.g., a problem of family of problems) where your bounds are better than current state-of-the-art bounds.

**Audience:**

I believe this work would have a very limited audience within the online learning theory community. I explain this in more detail below.

**Claims And Evidence:**

One reviewer had concerns about a claim that came up in the author's response to a review. Specifically, the authors mentioned that they can obtain another version of Theorem 3.1 (by using $T$ iterations of Construct) to allow Algorithm 3 to work for general convex functions rather than only strongly convex functions. However, without a proof of how this would work, it isn't possible to verify correctness. In addition, the comparison to the work of Zhao and Zhang seems to have issues, which I think the authors did acknowledge. That said, no reviewers questioned the correctness of the results that appear in the submitted manuscript.